# Conformational dynamics and asymmetry in multimodal inhibition of membrane-bound pyrophosphatases

Jianing Liu[1†], Anokhi Shah[2,3†], Xinyu Liu[2,3†], Joshua L Wort[2,3†], Yue Ma[2,3†], Katie Hardman[4], Niklas G Johansson[5], Orquidea Ribeiro[1], Adam Brookfield[6], Alice Bowen[6], Jari Yli-Kauhaluoma[5], Henri Xhaard[5], Lars JC Jeuken[7], Adrian Goldman[1*], Christos Pliotas[2,3*], Keni Vidilaseris[1*]

[1]Research Program in Molecular and Integrative Biosciences, University of Helsinki, Helsinki, Finland; [2]BioEmPiRe Centre for Structural Biological EPR Spectroscopy, School of Biological Sciences, Faculty of Biology, Medicine and Health, University of Manchester, Manchester, United Kingdom; [3]Manchester Institute of Biotechnology, University of Manchester, Manchester, United Kingdom; [4]Astbury Centre for Structural Molecular Biology, School of Biomedical Sciences, University of Leeds, Leeds, United Kingdom; [5]Drug Research Program, Division of Pharmaceutical Chemistry and Technology, Faculty of Pharmacy, University of Helsinki, Helsinki, Finland; [6]The National Research Facility for Electron Paramagnetic Resonance, The Photon Science Institute and The Department of Chemistry, University of Manchester, Manchester, United Kingdom; [7]Leiden Institute of Chemistry, University Leiden, Leiden, Netherlands

*For correspondence:
adrian.goldman@helsinki.fi (AG);
christos.pliotas@manchester.ac.uk (CP);
keni.vidilaseris@helsinki.fi (KV)

[†]These authors contributed equally to this work

Competing interest: The authors declare that no competing interests exist.

## eLife Assessment

This **important** study uncovers the mechanism of inhibition of a membrane pyrophosphatase by non-hydrolyzable phosphonate substrate analogs. **Convincing** crystallography, EPR spectroscopy, and functional measurements support the presence of a distinct conformational equilibrium of TmPPase in solution, and further supports the notion of asymmetric inhibitor binding at the active site, while maintaining a symmetric conformation at the periplasmic interface.

**Abstract** Membrane-bound pyrophosphatases (mPPases) are homodimeric proteins that hydrolyse pyrophosphate and pump $H^+/Na^+$ across membranes. They are crucial for the virulence of protist pathogens, making them attractive drug targets. In this study, we investigate the inhibitory effects of seven distinct bisphosphonates against *Thermotoga maritima* mPPase to explore their mode of action and assist in future small molecule inhibitor development. We solved two structures of mPPase bound to the inhibitors in the enzyme active sites and probed the conformational dynamics of mPPase under multiple inhibitors and functionally relevant conditions by double electron-electron resonance (DEER) spectroscopy. We found that mPPase adopts distinct conformational equilibria in solution in the presence of different inhibitors, including states consistent with asymmetric binding in the active site (closed-open), but a symmetric apo-like conformation on the periplasmic side (open-open). Combined with solid-supported membrane-based electrophysiology recordings, this revealed that during catalysis, one monomer of the dimer remains open, and $Na^+$ can only be pumped in a closed state. These results further support symmetry-breaking across the membrane, consistent with half-of-the-sites-reactivity.

## Introduction

Membrane-bound pyrophosphatases (mPPases) facilitate the transport of protons and/or sodium ions across membranes while catalysing the breakdown of pyrophosphate ($PP_i$), a by-product generated in various cellular synthetic reactions – into inorganic phosphate ($P_i$). These enzymes are found in plants, certain species of bacteria, protist parasites, and archaea, but are absent from multicellular animals (*Baltscheffsky et al., 1966*; *Karlsson, 1975*; *Rea et al., 1992*; *Baykov et al., 2013*; *Serrano et al., 2007*). Within these organisms, mPPases are essential for cell survival under diverse stress conditions such as osmotic stress, mineral deficiency, and extreme temperature (*Kajander et al., 2013*). Based on their potassium dependency, mPPases are divided into two families: $K^+$-dependent and $K^+$-independent. While $K^+$-independent mPPases all transport $H^+$, $K^+$-dependent mPPases can transport $H^+$, $Na^+$, or both (*Baykov et al., 2013*).

Currently, mPPase structures have been reported from three different organisms: *Vigna radiata* (VrPPase), *Thermotoga maritima* (TmPPase), and, most recently, a structure from *Pyrobaculum aerophilum* (PaPPase) in complex with imidodiphosphate (IDP) (*Strauss et al., 2024*). For TmPPase, several different structural states have been determined, including the resting-state (TmPPase:Ca:Mg)

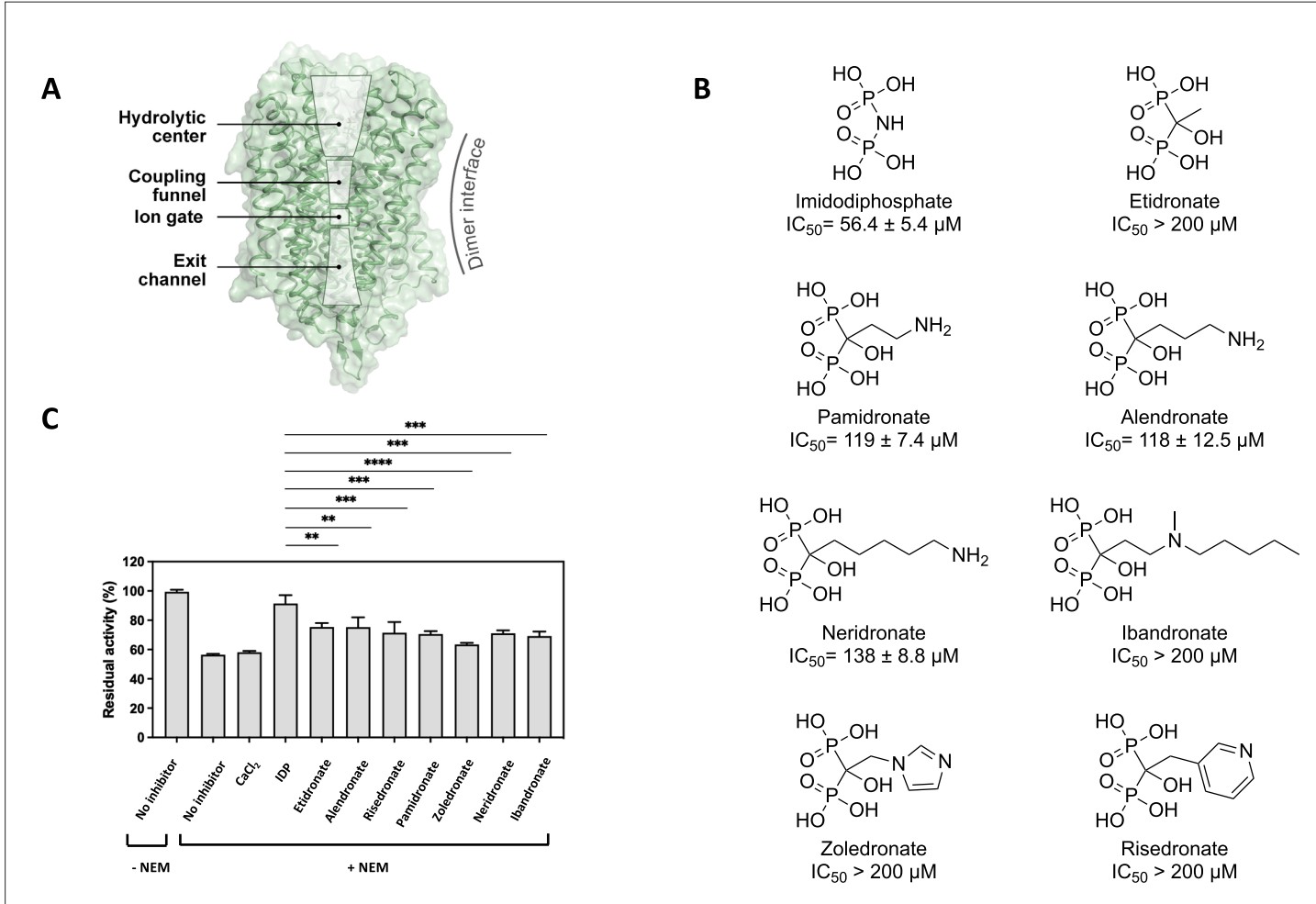

**Figure 1.** Inhibition effect of bisphosphonates on TmPPase. (**A**) Overall structure of the monomer TmPPase structure, consisting of hydrolytic centre, coupling funnel, ion gate, and exit channel. (**B**) Chemical structure of imidodiphosphate (IDP) and bisphosphonates and their inhibition activity against TmPPase. (**C**) Activity of TmPPase modified with NEM (*N*-ethylmaleimide) in the presence of $MgCl_2$ (2.5 mM) and NaCl (20 mM) after incubation with $Ca^{2+}$ (2 mM) and bisphosphonates (0.5 mM). Measurement was done in triplicate.

The online version of this article includes the following figure supplement(s) for figure 1:

**Figure supplement 1.** Inhibition of TmPPase by bisphosphonates.

**Figure supplement 2.** Top view of accessible cysteines for *N*-ethyl maleimide (NEM) modification.

(*Kellosalo et al., 2012*), with two phosphates bound (TmPPase:2P$_i$) (*Kellosalo et al., 2012*), IDP bound (TmPPase:IDP) (*Li et al., 2016*), IDP and *N*-[(2-amino-6-benzothiazolyl)methyl]-1*H*-indole-2-carboxamide (ATC) bound (TmPPase:IDP:ATC) (*Vidilaseris et al., 2019b*), phosphate analogue (WO$_4$)-bound (TmPPase:WO$_4$) (*Li et al., 2016*), and time-resolved X-ray diffraction structures (with and without substrate/product bound) showing structural asymmetry (*Strauss et al., 2024*). Similarly, VrPPase has been solved in multiple states, including IDP-bound (VrPPase:IDP) (*Lin et al., 2012*), single phosphate-bound (VrPPase:P$_i$) (*Li et al., 2016*), two phosphates bound (VrPPase:2P$_i$), and different mutations at the hydrophobic gate (*Tsai et al., 2019*). These structures show that mPPases are homodimeric enzymes, with each monomer consisting of 16 and, as found in sequence databases, occasionally 17 transmembrane helices (TMHs), organised into two concentric rings: the inner ring (TMH5–6, 11–12, and 15–16) and the outer ring (TMH1–4, 7–10, and 13–14). Each monomer consists of four regions: a hydrolytic centre, a coupling funnel, an ion gate, and an exit channel (*Tsai et al., 2014*; *Figure 1A*). To simplify residue comparison between mPPases, we employ the residue numbering scheme X$\Sigma^{Y,Z}$ (superscripts refer to Ballesteros-Weinstein numbering; *Ballesteros and Weinstein, 1995*), where X represents the amino acid, $\Sigma$ denotes the amino acid position in TmPPase, Y indicates the helix number, and Z specifies the offset of amino acid positions within the centrally conserved residues of the helix (*Tsai et al., 2014*).

mPPases are a promising drug target for treating diseases caused by parasitic protists, such as malaria and leishmaniasis (*Vidilaseris et al., 2019b*; *Johansson et al., 2021*; *Johansson et al., 2020*; *Shah et al., 2016*). Among the currently available compounds, ATC demonstrates the most effective inhibitory activity against TmPPase (*Vidilaseris et al., 2019b*). ATC is bound to a region near the enzyme exit channel of one subunit, which induces structural asymmetry in the mPPase dimer (*Vidilaseris et al., 2019b*). Functional asymmetry in K$^+$-dependent mPPases has also previously been shown by *Artukka et al., 2018*. *Anashkin et al., 2021*, further supported this hypothesis by analysing the inhibition of *Desulfitobacterium hafniense* mPPase using three non-hydrolysable PP$_i$ analogues (IDP, etidronate [ETD], and aminomethane bisphosphonate). Bisphosphonates, such as risedronate (RSD) and pamidronate (PAM), serve as primary drugs currently used to combat osteoclast-mediated bone loss (*Drake et al., 2008*). Unlike IDP, which contains a P-N-P bond, bisphosphonates have a P-C-P bond, with its central carbon can accommodate up to two substituents, allowing a large compound variability. Therefore, understanding their inhibition mechanism on mPPases is crucial for developing future small molecule inhibitors.

Our previous work on serial time-resolved X-ray crystallography and electrometric studies on TmPPase directly observed structural asymmetry, where two monomers are in different states during PP$_i$ hydrolysis upon the addition of substrate and Na$^+$, supporting a 'pumping-before-hydrolysis' energy coupling model (*Strauss et al., 2024*). However, except for the allosteric inhibitor ATC, which binds to a region near the exit channel, crystal structures of TmPPase bound to inhibitors at the active site are symmetric. To probe the proposed asymmetry caused by the inhibitor (and substrate) binding in solution, we employed double electron-electron resonance (DEER), also known as pulsed electron double resonance (PELDOR) spectroscopy. This method relies on the introduction of paramagnetic spin labels at selected protein residues, allowing for precise determination of electron-electron dipolar couplings and subsequently, inter-spin distances (*Jeschke, 2018*; *Schiemann and Prisner, 2007*; *Goldfarb, 2022*), making it a powerful tool for probing the conformation and dynamics of integral membrane proteins (*McHaourab et al., 2011*; *Bordignon et al., 2019*; *Hartley et al., 2020*; *Pliotas et al., 2012*; *Shah et al., 2025*), including ion channels, transporters, outer membrane proteins, and receptors in their native environments (*Kapsalis et al., 2019*; *Kapsalis et al., 2020*; *Gopinath et al., 2024*; *Galazzo et al., 2022*; *Thaker et al., 2022*; *Wingler et al., 2019*; *Haysom et al., 2023*). As an ensemble technique, DEER can probe the presence of multiple conformational species, including lowly populated protein states, which are key to protein function (*Beck et al., 2024*; *Lane et al., 2024*; *Zhao et al., 2024*). Here, we solved two TmPPase structures in complex with ETD and zoledronate (ZLD) and monitored their conformational ensemble using DEER spectroscopy in solution. Overall, bisphosphonates can trigger conformational changes in the active site and near the exit channel of TmPPase in an asymmetric mode and under certain inhibitor-bound conditions; the DEER data are consistent with interspin distances predicted from an open/closed asymmetric model and correlate with the corresponding X-ray structures. This, along with our electrometric studies detecting the Na$^+$ signal across the membrane, further suggests that ion pumping requires a fully closed state of one

TmPPase monomer, supporting symmetry-breaking across the membrane, consistent with half-of-the-sites-reactivity (*Strauss et al., 2024*).

## Results

### Bisphosphonates are weaker TmPPase inhibitors than IDP

Bisphosphonates have been shown to inhibit mPPases (*Anashkin et al., 2021*; *Gordon-Weeks et al., 1999*). To understand their binding mechanism to TmPPase, we first assessed the binding ability of seven distinct bisphosphonates to TmPPase by testing their inhibitory activity using the molybdenum blue assay (*Figure 1B*), with IDP ($IC_{50}$=56±5 µM) as a positive control (*Vidilaseris et al., 2019a*). Of the compounds tested, all the straight-chain primary amines (PAM, alendronate [ALE], and neridronate) had similar $IC_{50}$s, ranging from 117 to 138 µM (p=0.06). Substituting the –NH– of IDP with the –CCH$_3$(OH)– of ETD resulted in a weaker $IC_{50}$ (>200 µM). Similarly, branched aliphatic and aromatic bisphosphonates (ibandronate [IBD], ZLD, and RSD) also showed weaker inhibition ($IC_{50}$ >200 µM) (*Figure 1B*, *Figure 1—figure supplement 1*).

To confirm that the binding of bisphosphonates to TmPPase induces conformational changes in the protein structure, we incubated the enzyme with the inhibitors and performed an *N*-ethyl maleimide (NEM) modification assay (*Yamagata and Iwama, 1999*). NEM covalently binds to exposed cysteine residues of the protein, forming a carbon-sulphur bond that can inhibit the protein activity if the residue is essential (*Kellosalo et al., 2012*). The binding of IDP has been reported to prevent the NEM modification of cysteine by reducing cysteine accessibility, thereby preserving TmPPase activity (*Kellosalo et al., 2012*). In the absence of inhibitors, NEM modification resulted in a decrease in TmPPase activity by approximately 40% (*Figure 1C*), similar to the activity reduction observed with $CaCl_2$, an inhibitor that binds to the open form of TmPPase (*Kellosalo et al., 2012*). Upon the addition of IDP, TmPPase adopts a closed conformation, rendering it resistant to NEM modification (*Kellosalo et al., 2012*; *Figure 1—figure supplement 2*); consequently, the enzyme remains largely unaffected by NEM. Although not as effective as IDP, all bisphosphonates prevent NEM modification to a comparable extent (*Figure 1C*).

### TmPPase structures in complex with bisphosphonate inhibitors

To decipher the structural basis of bisphosphonates inhibition and their binding to TmPPase, we decided to solve their structures since all the bisphosphonates bound to TmPPase despite not being isosteres of PP$_i$ (*Figure 1*). We obtained protein crystals for all the inhibitors, but they diffracted weakly, except for TmPPase in complex with ETD (TmPPase:ETD) and ZLD (TmPPase:ZLD), which diffracted to resolutions of 3.2 and 3.3 Å, respectively. TmPPase:ETD crystallised in the presence of $Ca^{2+}$, which is a well-known mPPase inhibitor (*Kellosalo et al., 2012*), while TmPPase:ZLD crystallised without $Ca^{2+}$. Both datasets were anisotropic as analysed using the STARANISO server (*Tickle et al., 2016*; *Supplementary file 1*). We solved both structures by molecular replacement using the resting-state structure (PDB ID: 4AV3) as the search model for TmPPase:ETD and the closed IDP-bound structure (PDB ID: 5LZQ) for TmPPase:ZLD. There were two molecules in the asymmetric unit for TmPPase:ETD and four for TmPPase:ZLD.

In the initial round of the refinement for the TmPPase:ETD structure, both chains displayed positive ($F_o$–$F_c$) density at 3σ in their hydrolytic centres that could accommodate ETD (*Figure 2—figure supplement 1A and B*, upper left panel). We also observed extra density that corresponds to a calcium ion in the resting-state structure (*Kellosalo et al., 2012*; *Figure 2—figure supplement 1A and B*, upper left panel). Due to the high $Ca^{2+}$ concentration (0.2 M) in the crystallisation condition, we placed the same ion at this position. After placing $Mg^{2+}$ ions and water molecules in the difference density peaks, further rounds of refinement provided us with a reasonable $2mF_o$–$F_c$ density map of the active site of both monomers (*Figure 2—figure supplement 1*, right panel) and the Polder (omit) maps indicate a good fit of the compound to the density (*Figure 2—figure supplement 1*, bottom left panel). Finally, the TmPPase:ETD structure was refined to an average resolution of 3.2 Å (h=3.1 Å, k=3.6 Å, l=4.3 Å) with the final $R_{work}$/$R_{free}$ of 27.2%/31.0% (*Supplementary file 2*).

Similarly, the initial refinement of TmPPase:ZLD revealed positive ($F_o$–$F_c$) density at 3σ that could accommodate ZLD in all four chains in the asymmetric unit (*Figure 3—figure supplement 1A–D*, upper left panel). After placing $Mg^{2+}$ ions and water molecules into the difference density peaks,

further rounds of refinement provided us with a $2mF_o–F_c$ density map of the active site for all monomers (*Figure 3—figure supplement 1A–D*, right panel) and was validated by Polder (omit) maps (*Figure 3—figure supplement 1A–D*, bottom left panel). The final refinement shows that the TmPPase:ZLD structure has an average resolution of 3.3 Å (h=4.5 Å, k=4.2 Å, l=3.2 Å) with a final $R_{work}/R_{free}$ of 25.9%/30.4% (*Supplementary file 1*).

## Asymmetry in the TmPPase complex with etidronate

Unlike the fully open TmPPase:Ca:Mg structure (PDB ID: 4AV3), there was additional density above the hydrolytic centre in both chains that could be fitted with several residues of loop5–6 (*Figure 2A*, *Figure 2—figure supplement 2*). This left eight residues (V208[5.67]–L215[5.74]) in loop5–6 of chain A and three residues (L213[5.72]–L215[5.74]) in chain B unmodeled due to the lack of extra density. In the IDP-bound structure, these loops interact with IDP and form a tightly packed structured lid over the active site. However, in the TmPPase:ETD structure, despite interacting with ETD in both chains, these loops are positioned slightly above the active site (*Figure 2B–D*), with loop5–6 of chain A extending more toward the centre compared to loop5–6 of chain B (*Figure 2A*).

Structural alignment between chain A and B of TmPPase:ETD yields a root mean square deviation (RMSD) per Cα of 1.44 Å, approximately four times higher than the RMSD between chain A and B in the resting state (RMSD/Cα=0.39 Å) (*Supplementary file 2*). Despite the overall structural similarity, further comparison between the monomers of TmPPase:ETD and those in the resting state revealed that chain B of TmPPase:ETD differs most from TmPPase:ETD chain A and from both chains in the resting-state structure (*Supplementary file 2*). Notably, there are clear differences on the cytoplasmic (hydrolytic) side between the monomers of TmPPase:ETD; chain B adopts a slightly more constricted conformation than chain A (*Figure 2A*). Bendix analysis (*Dahl et al., 2012*) showed that three (TMH11, 12, and 15) out of six inner ring helices of chain B are more curved on the cytoplasmic side, bending towards the active site (*Figure 2—figure supplement 3*). Besides that, the loops on the cytoplasmic side of chain B (loops11–12, 13–14, and 15–16) appear to be more flexible, as indicated by more unresolved residues, compared to those in chain A (*Figure 2A*). These observations suggest structural asymmetry between chains A and B in the TmPPase:ETD structure.

The structural asymmetry arises because the binding of $ETD_A$ to monomer A induced conformational changes in monomer B, thereby affecting the binding pose of $ETD_B$ in monomer B. ETD comprises two phosphonate groups separated by a central carbon bonded to a hydroxyl group. Compared to the IDP location in the IDP-bound structure, ETDs are positioned above the IDP site, with the lower phosphonate group of ETDs located in the position of the upper (leaving-group) phosphonate group of IDP (*Figure 2B*). However, the upper phosphonate group of ETDs in chains A ($ETD_A$) and B ($ETD_B$) is distinctly positioned; $ETD_A$ is tilted approximately by 35.9° relative to the IDP orientation, while $ETD_B$ is parallel to the IDP orientation (*Figure 2B*). The lower phosphonate group position remains the same for both ETDs (*Figure 2B*). As a result, loop5–6 of the two monomers is oriented differently. In chain A, this loop protrudes towards the active centre and E217[5.76] interacts with $ETD_A$ via an $Mg^{2+}$ ion, while in chain B, the loop is more constricted and interacts with $ETD_B$ via D218[5.77], also mediated by an $Mg^{2+}$ ion (*Figure 2B, C, and D*). Furthermore, $ETD_A$ and $ETD_B$ interact with the active site via different residues (*Figure 2C and D*). D465[11.57], D488[12.39], and N492[12.43] in TMH11 and TMH12 were involved in the interaction with $ETD_B$ via a water molecule. Consequently, these two TMHs undergo slight inward movement, resulting in a more constricted conformation of chain B. Exchanging the ETD positions between the two protomers generated corresponding positive and negative difference electron density peaks, confirming distinct conformations of ETD within each protomer (*Figure 2—figure supplement 1C*). Nonetheless, the methyl group of ETDs in both chains points towards TMH12 (*Figure 2C and D*), which might prevent complete closure of the hydrolytic centre and downward motion of TMH12.

## Structural distinction between zoledronate and IDP-bound TmPPase

In contrast to the TmPPase:ETD structure, the TmPPase:ZLD structure adopts a partially closed conformation. The overall structure is more similar to the IDP-bound structure (RMSD/Cα of 0.760 Å) than the resting-state structure (RMSD/Cα of 2.32 Å) (*Figure 4—figure supplement 1*). However, compared to the IDP-bound structure, the TmPPase:ZLD structure exhibits noticeable movements in three of six inner ring helices (TMH11, 12, and 15) and seven of ten outer ring helices (TMH1–4 and 7–9). These

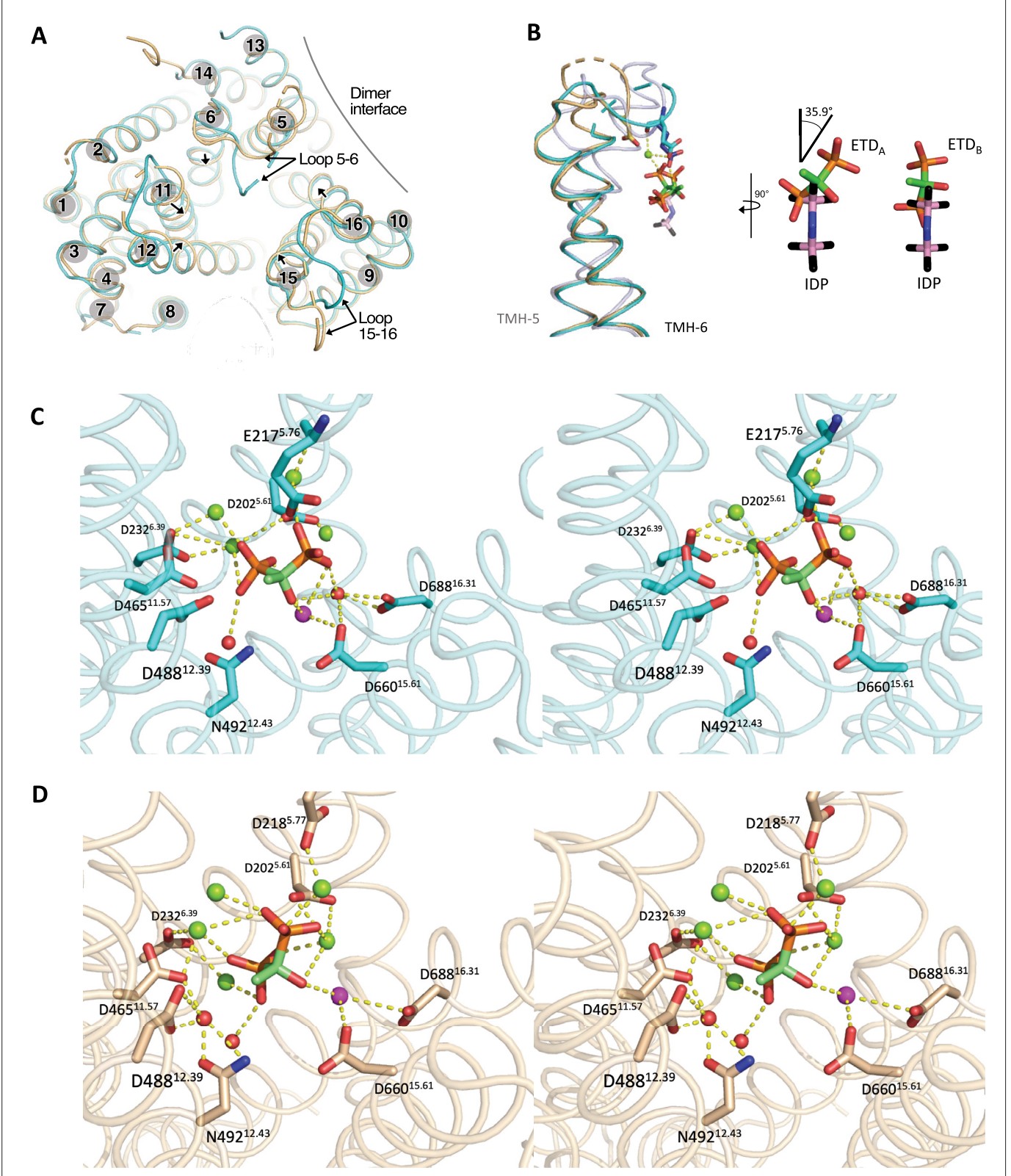

**Figure 2.** Structural asymmetry in the dimer active site of TmPPase:ETD complex. (**A**) Top view of the superposition of chain A (cyan) and chain B (wheat) showing the relative movements (black arrow) of helices. (**B**) Side view of the superposition of TMH5 and TMH6 in TmPPase:ETDs (chain A [cyan] and chain B [wheat]) and TmPPase:IDP (light blue; PDB: 5LZQ) showing the movement of loop5–6 and reorientation of ETD$_A$, ETD$_B$, and IDP. The yellow dashed line shows the interaction of E217$^{5.76}$ of loop5–6 in chain A with ETD$_A$ and IDP, and D218$^{5.77}$ of loop5–6 in chain B with ETD$_B$. Close-up view of IDP

*Figure 2 continued on next page*

*Figure 2 continued*

superposed with ETD$_A$ and ETD$_B$. (**C**) Stereo representation (wall-eyed view) of residues in the active site with ETD$_A$ coordinated (dashed lines), Ca$^{2+}$ (pink sphere), and nucleophilic water (red spheres) in an Mg$^{2+}$ metal cage (green spheres). (**D**) Stereo representation (wall-eyed view) of residues in the active site with ETD$_B$ coordinated (dashed lines), Ca$^{2+}$ (pink sphere), and water (red spheres) in an Mg$^{2+}$ metal cage (green spheres).

The online version of this article includes the following figure supplement(s) for figure 2:

**Figure supplement 1.** Electron density maps of etidronate (ETD) at the active sites.

**Figure supplement 2.** Electron density maps of loop5–6 in the TmPPase:ETD structure.

**Figure supplement 3.** Helix curvature comparison between chain A and chain B of the TmPPase:ETD structure.

movements extend outwards from the hydrolytic centre (*Figure 3A*), leaving it only partially closed. A cross-sectional view confirms this observation, showing the tunnel extending from the hydrolytic centre to the enzyme surface, unlike in the IDP-bound structure, where it is closed (*Figure 3B and C*). This is because ZLD is sterically bulkier than IDP due to the presence of the heteroaryl group, which points towards TMH11, 12, and 15 on the cytoplasmic side.

Although the hydrolytic centre of TmPPase:ZLD is more open, the coordination of the Mg$_4$ZLD complex with the active site residues closely resembles that of Mg$_5$IDP in the IDP-bound structure (*Figure 3E and F*). ZLD is nonetheless positioned about 1.0 Å above IDP (*Figure 3D*) because the steric bulk prevents it from sitting deeper into the hydrolytic centre. However, unlike the IDP-bound structure, and even though the arrangement of TMHs in the ion gate is almost identical, we did not observe any density for a Na$^+$ in the TmPPase:ZLD structure despite its higher resolution (i.e. 3.26 Å compared to 3.5 Å for the IDP-bound structure) (*Figure 3—figure supplement 2C*).

## Probing the solution-state conformational ensemble and dynamics of TmPPase by DEER spectroscopy

The X-ray structures of TmPPase with the different inhibitors bound to the active site show either a closed (TmPPase:IDP; *Li et al., 2016*), resting (TmPPase:Ca; *Kellosalo et al., 2012*), or asymmetric (TmPPase:ETD, *Figure 2*) conformation. The asymmetric structure of the TmPPase with ETD is similar to that observed in our recent time-resolved study (*Strauss et al., 2024*). To probe the TmPPase conformational ensemble in solution under various inhibitor-bound conditions, we employed DEER spectroscopy. We selected three distinct sites (periplasmic side, S525; cytoplasmic side, C599; cytoplasmic side loop region, T211) on TmPPase, which were selectively spin-labelled with 2,5-dihydro-2,2,5,5-tetramethyl-3-[[(methylsulfonyl)thio]methyl]-1*H*-pyrrol-1-yloxy (MTSSL, modification denoted as R1 hereafter) to enable the measurement of interspin distances between the spin-labelled residue pairs. The selected sites were designed to capture the coupled gating transitions and conformational changes occurring on either side of the membrane (*Figure 4A*, *Figure 4—figure supplement 1A*) without interfering with the activity of TmPPase. We achieved high mPPase spin labelling efficiency with no free (i.e. unbound or non-specifically bound) MTSSL being present, as evidenced by the continuous wave electron paramagnetic resonance (CW-EPR) spectra recorded at room temperature (*Figure 4—figure supplement 2*). CW-EPR spectra, which relate to the rotational correlation time, indicated that the spin label mobility increased sequentially from C599R1 to S525R1 and further to T211R1 across several tested conditions (*apo*, +Ca, +Ca/ETD, +ETD, +IDP, etc.). This mobility trend aligns with the location of T211R1 on an exposed loop, which explains its higher mobility, whereas spin labelling of the more buried C599R1 required the addition of Ca$^{2+}$ during sample preparation to induce partial structural opening. Unlike DEER, which provides insights into the long-range conformational changes of membrane proteins, CW-EPR offers information on the local environment of the spin label. The results show no significant difference in the local environment between the *apo* and inhibitor-bound state(s).

In addition, we generated in silico predictions of distance distributions for the three sites (S525R1, T211R1, and C599R1) using MtsslWizard (*Hagelueken et al., 2012*) and ChiLife (*Tessmer and Stoll, 2023*), based on the X-ray structures of TmPPase bound to different molecules (*Figure 4*, *Figure 4—figure supplement 1*). In the case of T211R1, the X-ray electron density in loop$_A$5–6 of the TmPPase:ETD (residues V208$^{5.67}$-L215$^{5.74}$: VGKTELNL) and TmPPase:Ca (residues T211$^{5.70}$-R221$^{6.28}$: TELNLPEDDPR) structures is missing, suggesting a highly dynamic or disordered state for this region. We therefore modelled this region using the Rosetta server (*Song et al., 2013*) and used that to generate in silico

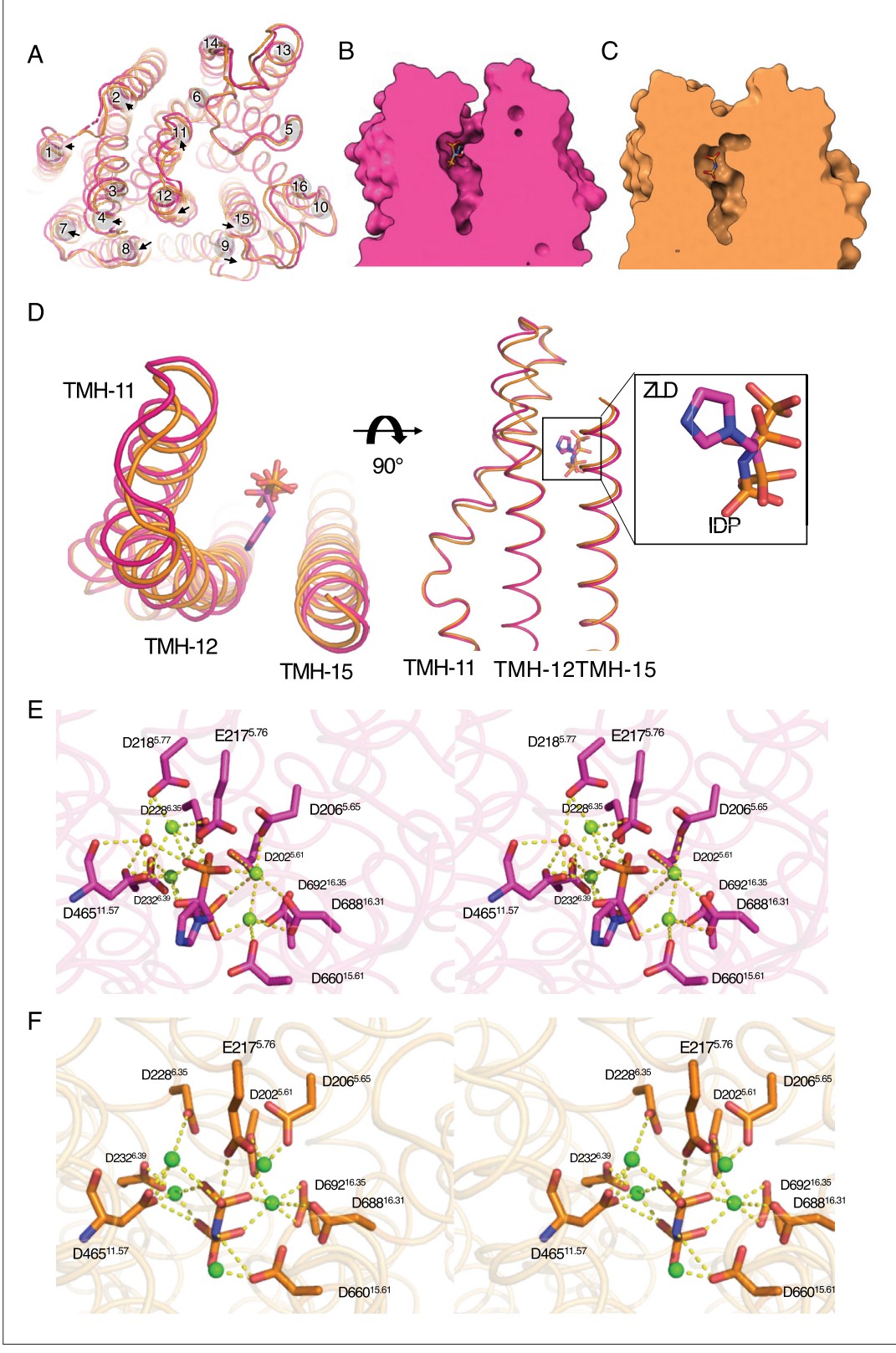

**Figure 3.** Comparison of the TmPPase:ZLD and TmPPase:IDP structures in the active site. (**A**) Top view of the superposition of TmPPase:ZLD (chain A, pink) and TmPPase:IDP (chain A, orange) (PDB: 5LZQ). Helix movements are indicated by a black arrow. (**B**) Cross-sectional view of the active site in TmPPase:ZLD. (**C**) Cross-sectional side view of the active site in TmPPase:IDP. (**D**) Top view of the superposition of TMH11, TMH12, and TMH15 in

*Figure 3 continued on next page*

*Figure 3 continued*

TmPPase:ZLD and TmPPase:IDP showed the movement of the hydrolytic centre and the orientation of ZLD and IDP. (**E**) Stereo representation (wall-eyed view) showing the coordination of key residues in the active site with ZLD (dash line) and water (red sphere) in an $Mg^{2+}$ metal cage (green spheres). (**F**) Stereo representation (wall-eyed view) showing the coordination of key residues in the active site with IDP (dash line) in an $Mg^{2+}$ metal cage (green spheres).

The online version of this article includes the following figure supplement(s) for figure 3:

**Figure supplement 1.** Electron density maps of zoledronate (ZLD) at the active sites.

**Figure supplement 2.** Ion gates of TmPPase structures.

---

distance distributions. These were overlaid with the experimentally derived DEER distance distributions (*Figure 4D, G*, *Figure 4—figure supplement 1D*) for comparison. All T211R1 distance distributions were broad, consistent with the increased spin label mobility observed by CW-EPR, and the highly dynamic nature of these loop regions (*Shah et al., 2017*). Owing to the featureless raw DEER data recorded for 211R1 (*Figure 4—figure supplement 1*), and broad distance distributions, we refrain from interpreting equilibria shifts based on this mutant. On the other hand, TmPPase dimers labelled at positions S525R1 and C599R1, located on opposite sides of the membrane, yielded high-quality DEER traces. Under all eight conditions tested (*apo*, +Ca, +Ca/ETD, +ETD, +IDP, +ZLD, +PAM, +ALE) for each site, strong dipolar oscillations were observed in the raw DEER data yielding robust distance distributions (*Figure 4B and E*). This indicates that the modal distance shifts observed within the TmPPase ensemble are highly reliable. Both DeerAnalysis2022 (*Jeschke et al., 2006*) and ComparativeDeerAnalyser 2.0 (*Fábregas Ibáñez et al., 2020*) were used for background correction and regularisation of the dipolar traces, and their resulting distance distributions were in good agreement (*Figure 4C, F*, *Figure 4—figure supplement 3*).

The separation of the S525R1 pair in the *apo* state (with no $Ca^{2+}$ or inhibitor added) is broad with a modal distance of 3.8 nm (full width at half-maximum [FWHM]=1.4 nm; σ=0.60 nm) (*Figure 4D*). In the presence of $Ca^{2+}$, the distance distribution is consistent with the predicted distances derived from the TmPPase:Ca structure, and the modal distance decreases (3.6 nm; FWHM = 1.0 nm; σ=0.43 nm). In the presence of both $Ca^{2+}$ and ETD (+Ca/ETD), we observe a similar modal distance (3.7 nm; FWHM = 1.2 nm; σ=0.51 nm) to that of the apo and $Ca^{2+}$ conditions, and the distribution is consistent with the predicted distance for the TmPPase:ETD structure (which corresponds to the +Ca/ETD condition). Furthermore, in the presence of ETD but no $Ca^{2+}$, the modal distance between the S525R1 pair on the different monomers increases to 3.9 nm (FWHM = 1.4 nm; σ=0.60 nm). Although these shifts are relatively small, under favourable conditions, DEER has the resolution to discriminate minute helical motions (*Pliotas et al., 2012*; *Peter et al., 2022*; *Klose et al., 2021*; *Pliotas, 2017*; *Pliotas et al., 2015*). The concerted nature of the modal distance shifts with respect to multiple different conditions at a single labelling site strongly suggests that preferential rotamer orientations are not the cause.

Upon visual inspection of the time-domain data (*Figure 4C*), the first minimum of the dipolar oscillation, as indicated by the black dashed lines depicted for the *apo* state, shifts to shorter time (i.e. higher frequency; shorter distance) for the TmPPase+$Ca^{2+}$ condition, and to longer time (i.e. lower frequency; longer distance) for the TmPPase+ETD condition, recapitulating the trends observed in the distance domain. Interestingly, upon the addition of IDP, the resulting distribution has modal distance of 4.0 nm (FWHM = 1.4 nm, σ=0.60 nm); shorter than the predicted distance for the TmPPase:IDP structure (4.3 nm). Meanwhile, with the addition of PAM and ALE, the resulting distributions have modal distances (+PAM: modal distance = 4.1 nm, FWHM = 1.2 nm, σ=0.51 nm; +ALE: modal distance = 4.3 nm, FWHM = 1.3 nm, σ=0.55 nm) similar to the in silico distance distribution predicted from the TmPPase:IDP X-ray structure. In contrast, the addition of ZLD results in the shortest modal distance observed for the S525R1 pair, of 3.4 nm (FWHM = 1.2 nm, σ=0.51 nm). Remarkably, this differs substantially from the in silico distance distribution predicted from the X-ray structure of TmPPase:ZLD (4.3 nm), which is expected to be highly similar to that of TmPPase:IDP (RMSD/Cα=0.571 Å) (see Discussion).

For the C599R1 dimer, the modal distance observed for all distributions under the tested conditions is approximately 5.8 nm (*apo*: modal distance = 5.8 nm, FWHM = 0.80 nm, σ=0.34 nm; +Ca: modal distance = 6.0 nm, FWHM = 0.80 nm, σ=0.34 nm; +IDP: modal distance = 5.8 nm, FWHM = 1.2 nm, σ=0.51 nm; +ZLD: modal distance = 5.9 nm, FWHM = 0.80 nm, σ=0.34 nm; +ETD: modal

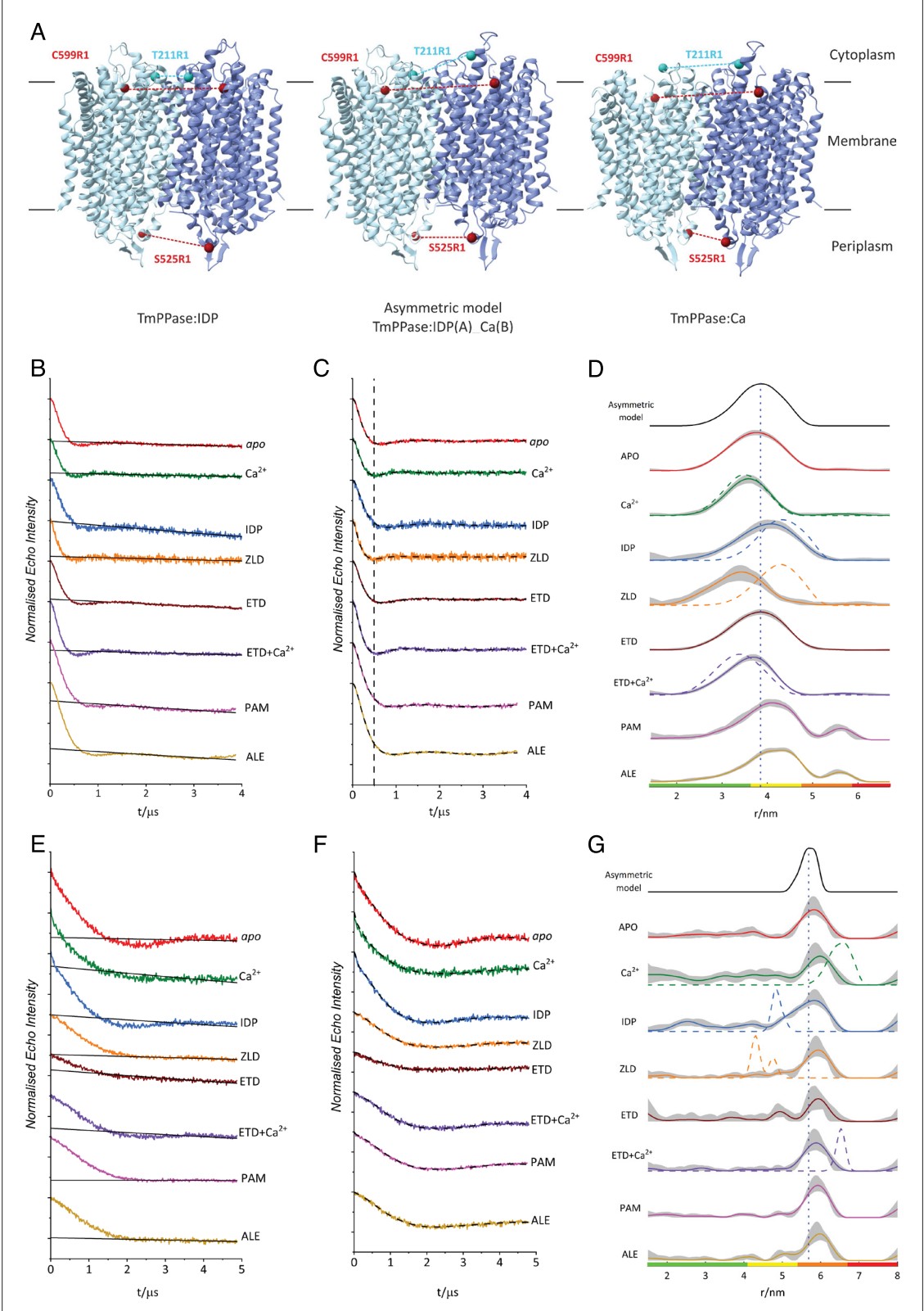

**Figure 4.** Double electron-electron resonance (DEER) distance distributions of TmPPase S525R1 and C599R1 under different conditions. (**A**) Symmetric structures (TmPPase:IDP [PDB: 5LZQ] and TmPPase:Ca [PDB: 4AV3]) and asymmetric model (TmPPase:IDP(A)_Ca(B)) of TmPPase. The sites mutated and labelled with MTSSL are shown as spheres, with T211R1 (cyan) and C599R1 (maroon) on the cytoplasmic side (top) and S525R1 (maroon) on the periplasmic side (bottom) of the membrane. Distances between spin pairs are indicated as dashed lines, consistent with sphere colouring. DEER data

*Figure 4 continued on next page*

*Figure 4 continued*

of T211R1 is shown in *Figure 4—figure supplement 1* EV8. (**B** and **E**) DEER raw data traces for S525R1 and C599R1, respectively. Each condition is labelled, and the raw data are colour-coded, with the background function indicated as solid black lines. (**C** and **F**) DEER background-corrected time-domain traces for S525R1 and C599R1, respectively. The vertical black dashed line represents the minimum of the first oscillation in the apo state and aids visualisation to highlight the shifts in the oscillation minimum under different conditions. (**D** and **G**) Distance distributions of S525R1 and C599R1, respectively. The in silico distance distribution corresponding to each spin pair modelled onto the asymmetric hybrid structure (TmPPase:IDP(A)_Ca(B)) is shown at the top as a solid black line, with the modal distance shown as a vertical dashed line. In silico predicted distance distributions for each condition, modelled using the solved structures (TmPPase:Ca, TmPPase:Ca:ETD [PDB 9G8K], TmPPase:ZLD [PDB 9G8J], and TmPPase:IDP) are presented as coloured dashed lines overlaying the experimental distributions. The shaded regions represent the 95% (2σ) confidence interval of the distributions, and the colour bars represent an assessment of the reliability of the distributions. The probability density within the green region indicates the mean distance, width, and peak shape are all reliable; the probability density within the yellow region indicates the mean distance and width are reliable; the probability density within the orange region indicates that the mean distance is reliable; the probability density within the red region indicates no quantitation is possible.

The online version of this article includes the following figure supplement(s) for figure 4:

**Figure supplement 1.** Double electron-electron resonance (DEER) distance distributions of TmPPase T211R1 under different conditions.

**Figure supplement 2.** Continuous wave electron paramagnetic resonance (CW-EPR) spectra of TmPPase T211R1, C599R1, and S525R1 under different conditions.

**Figure supplement 3.** ComparativeDeerAnalyzer (CDA) data of TmPPase S525R1, C599R1, and T211R1.

---

distance = 5.9 nm, FWHM = 0.70 nm, σ=0.30 nm; +ETD/Ca: modal distance = 5.9 nm, FWHM = 0.80 nm, σ=0.34 nm; +PAM: modal distance = 5.9 nm, FWHM = 0.70 nm, σ=0.30 nm; +ALE: modal distance = 6.0 nm, FWHM = 0.70 nm, σ=0.30 nm; *Figure 4G*); this is longer than the predicted 4.8 nm distance derived from the TmPPase:IDP structure – where both monomers are closed – but significantly shorter than the predicted 6.6 nm distance for the TmPPase:Ca and TmPPase:ETD structures, where both monomers are open. This deviation between prediction and experiment could be explained by the dimer adopting an asymmetric conformation under the physiological conditions used for DEER, with one monomer in a closed state and the other in an open state. To investigate the asymmetric arrangement between two TmPPase monomers, we combined chain A of the TmPPase:IDP structure with chain B of the TmPPase:Ca structure to generate an asymmetric model, termed TmPPase:IDP(A)_Ca(B). We refer to the conformation of the TmPPase:IDP structure as 'closed' at both sides, even for residues not in the active site, for residues as in the TmPPase:Ca structure as 'open' at both sides. Our asymmetric model has, for instance, S525(A) 'closed' but S525(B) 'open'. The asymmetric model predicts a distance distribution that agrees closely with the DEER data obtained for the majority of the eight conditions tested for both C599R1 and S525R1 pairs (*Figure 4D and G*). The distribution predicted by the asymmetric model also falls between the two conformational extremes (fully closed and fully open states) of TmPPase structures. To further delineate the best-fitting model of the S525R1 DEER data, particularly given their smaller range from 3.6 to 4.0 nm, which resembles both asymmetric (i.e. closed-open) and *apo* state (i.e. open-open) models, Bhattacharyya coefficients (*Bhattacharyya, 1946*) were calculated for the two models. The values are as follows: +Ca = 0.98 (*apo* model), 0.90 (asymmetric model); +IDP = 0.97 (apo model), 0.98 (asymmetric model); +ETD = 1.0 (*apo* model), 0.97 (asymmetric model); +ZLD = 0.95 (*apo* model), 0.84 (asymmetric model); +Ca/ETD = 0.98 (*apo* model), 0.91 (asymmetric model). It was not feasible to calculate these coefficients for the 525R1 +PAM and +ALE conditions, owing to being recorded on a different instrument, with a different x-axis, which was also the case for the C599R1 dataset. These coefficients for S525R1 indicate that the *apo* state (i.e. open-open) model describes the experimentally derived distributions better for +Ca, +Ca/ETD, +ETD, and +ZLD, whereas the asymmetric (i.e. closed-open) model better describes the experimental data for +IDP. Higher Bhattacharyya coefficient values (closer to unity) signify better overlap (here taken as a proxy for model agreement). The ramifications of these calculations are further elaborated in the discussion.

## Effect of ETD and ZLD on sodium transport of TmPPase

Previously, we showed that IDP can facilitate a single $Na^+$ pumping cycle without hydrolysis (*Strauss et al., 2024*). To investigate whether pumping also occurs in the presence of ETD and ZLD, we recorded electrometric data during $PP_i$ hydrolysis and after binding of IDP, ETD, and ZLD. In electrometric measurements, also known as solid-supported membrane-based electrophysiology (*Bazzone et al.,*

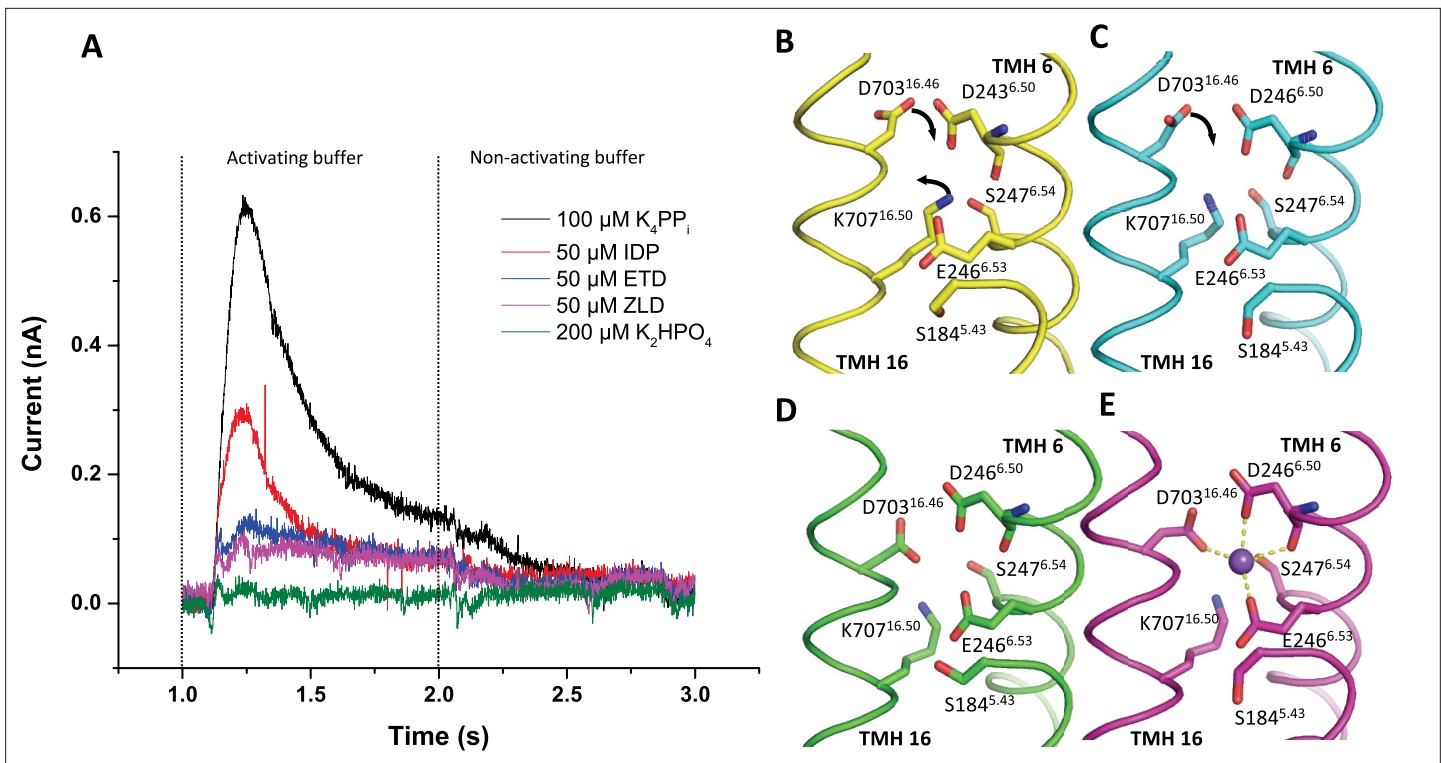

**Figure 5.** Transient currents of TmPPase Na$^+$ pumping and ion gate of TmPPase structures. (**A**) Curve of Na$^+$ pumping current triggered by 100 μM of K$_4$PP$_i$, 50 μM of IDP, 50 μM of etidronate (ETD), 50 μM of zoledronate (ZLD), and 200 μM of K$_2$HPO$_4$. The vertical black dashed line represents the addition of activating buffer and non-activating buffer. (**B–E**) Ion gate of TmPPase:Ca (yellow); TmPPase:ETD (cyan); TmPPase:ZLD (green); TmPPase:IDP (purple). The black arrows show the movement of residues of D703$^{16.46}$ and K707$^{16.50}$.

*2017*), a current signal is generated and recorded when Na$^+$ is transported across the membrane by the active reconstituted TmPPase. A maximal positive signal of 0.6±0.03 nA was detected within 0.15 ns (excluding instrument dead time) after the addition of 100 μM substrate K$_4$PP$_i$ (*Figure 5A*). Most of the signal decayed within 1 s after K$_4$PP$_i$ was added. Full signal recovery required several minutes before a repeat measurement could be performed on the same sensor. As expected, when 200 μM K$_2$HPO$_4$ was added as a negative control, there was no signal, indicating that no ion pumping had occurred. Replacing the substrate with IDP resulted in a signal about half that of K$_4$PP$_i$. However, in the presence of 50 μM ETD or 50 μM ZLD, the signals were barely detectable, indicating no Na$^+$ pumping was observed.

This observation is consistent with the DEER data described above and with the TmPPase:ETD and TmPPase:ZLD structures, where there is no density for Na$^+$ in the ion gate. Interestingly, in all solved TmPPase structures, Na$^+$ has been observed at the ion gate only in the IDP-bound structures (*Figure 5B–E*, *Figure 3—figure supplement 2*). In the IDP-bound structure, four key residues (D703$^{16.46}$, D243$^{6.50}$, S247$^{6.54}$, and E246$^{6.53}$) in the ion gate constitute the Na$^+$ binding site (*Figure 5E*). The formation of the site is driven by the downward motion of TMH16 (*Figure 3—figure supplement 2A*), transitioning from the resting state (TmPPase:Ca) to the closed state (TmPPase:IDP). The orientation of D703$^{16.46}$ of the TmPPase:ETD structure resembles the structure of TmPPase:Ca, rotated away from the Na$^+$ binding site, causing a loss of Na$^+$ binding (*Figure 5B and C*). In the TmPPase:ZLD structure, D703$^{16.46}$ and K707$^{16.50}$ are oriented relatively similarly to the Na$^+$ binding position in the TmPPase:IDP structure (*Figure 5D, E*, *Figure 3—figure supplement 2C*), but no Na$^+$ density was observed despite the higher resolution compared to the TmPPase:IDP structure (3.26 Å compared to 3.5 Å for the IDP-bound structure). This might be because the inhibitor restricts the complete closure of the active site and full constriction and downward movement of the inner helices (especially TMH12 and 16) (*Figure 3A–D*), which hinder the Na$^+$ pumping.

# Discussion

## Inhibition of TmPPase by bisphosphonates

The seven distinct bisphosphonates we tested exhibited varying levels of inhibition against TmPPase (*Figure 1B*). ETD and IBD exhibited higher $IC_{50}$ values compared to PAM, ALE, and NRD (p<0.0001) (*Figure 1B*). This is consistent with the $K_i$ of ETD on DhPPase (mPPase of *D. hafniense*), which is approximately 67 times higher than that of amino methylene diphosphonate, as measured by *Anashkin et al., 2021*; *Malinen et al., 2022*. The difference may be due to the introduction of an amino group in the side chain and its length (*Gordon-Weeks et al., 1999*). Substituting the hydrogen (in ETD) with the benzene ring (in ZLD and RSD) decreases inhibitory activity ($IC_{50}$ >200 µM). Nonetheless, these aromatic-containing compounds are still capable of preventing NEM modification on TmPPase (*Figure 1C*), as further supported by the solved structure of ZLD bound with TmPPase (see Discussion section below).

## Catalytic asymmetry in mPPase

Some evidence for asymmetry in mPPase gating has been shown previously by kinetic studies (*Strauss et al., 2024*; *Anashkin et al., 2021*) and captured in the time-resolved 600 and 3600 s structures of TmPPase:PP$_i$ (*Strauss et al., 2024*), where in both structures, one chain is in the open state (i.e. as in the *apo* structure) and the other is in the closed state (i.e. as in the IDP-bound structure). Our DEER data reveal clear differences in the binding of different inhibitors leading to a variety of open-closed states: IDP generates a closed-open state on both sides of the membrane, consistent with the presence of Na$^+$ in the ionic gate and pumping, while ETD and ZLD generate a closed-open state on the cytoplasmic side, but open-open states on the periplasmic side. These begin to explain the conformation changes upon substrate/inhibitor binding.

In our study, the TmPPase:ETD structure captured the asymmetric binding of ETD (*Figure 2*). Loop5–6, which interacts with ETD, moves inward to partially close the active site, but not as deeply as observed in the TmPPase:IDP structure (*Figure 2A and B*). Here, ETD is positioned above the hydrolytic centre (*Figure 2B*) and cannot descend further due to the presence of Ca$^{2+}$ in the active site (*Figure 2C and D*), similar to the TmPPase:Ca structure (PDB ID: 4AV3). Thus, although ETD induces partial closure of loop5–6 and provides some stabilisation, the overall arrangement of the inner and outer helices remains more like the open state rather than the fully closed state (*Figure 2A*).

The DEER data on C599R1 provided reliable DEER distance distributions, and across all eight conditions tested, supported an asymmetric binding mode of compounds to TmPPase at the cytoplasmic side (*Figure 4E–G*). The modal distance of 5.8 nm differs significantly from the predicted C599R1 modal distance in the TmPPase:Ca (6.8 nm) and TmPPase:IDP (4.8 nm) structures. The presence of a minor population at approximately 5 nm observed in the presence of IDP or ETD is consistent with the predicted distance for the TmPPase:IDP structure, where both monomers are in a fully closed conformation. The 5.8 nm major peak corresponds to the closed-open conformation for IDP and ETD. Consequently, the DEER data on the cytoplasmic side demonstrate an equilibrium of at least two states: a minor population with IDP/ETD bound to both active sites, leading to a fully closed conformation on the cytoplasmic side, and a major population with IDP/ETD only bound to one active site, yielding an asymmetric closed-open state. This corresponds to the observed mechanism of substrate inhibition (*Strauss et al., 2024*; *Vidilaseris et al., 2019b*), where binding to both active sites (i.e. closed-closed) decreases the activity of the enzyme in comparison with the half-occupied open-closed state. Under the condition tested, we did not observe ZLD, PAM, and ALE bound to both active sites, probably due to the bulkiness of the compounds.

We cannot completely rule out the possibility that the monomers adopt a metastable intermediate state: in such a case, we would expect the distance changes reported by DEER to be symmetric across both membrane sides. However, we observe symmetry breaking between the cytoplasmic and periplasmic TmPPase sites. Indeed, DEER data yield distance distributions similar to that of the hybrid asymmetric structure under all conditions (*apo*, +Ca, +Ca/ETD, +ETD, +ZLD, +IDP, +PAM, +ALE). The distance distribution for S525R1 (loop12–13) in the exit channel changes more between different conditions than C599R1 on the cytoplasmic side (*Figure 4*). Under +Ca and +Ca/ETD conditions, its distance distribution remains largely unchanged, with a mean distance of ~3.5–3.7 nm (*Figure 4D*), which is consistent with the predicted distance derived from their corresponding crystal structures.

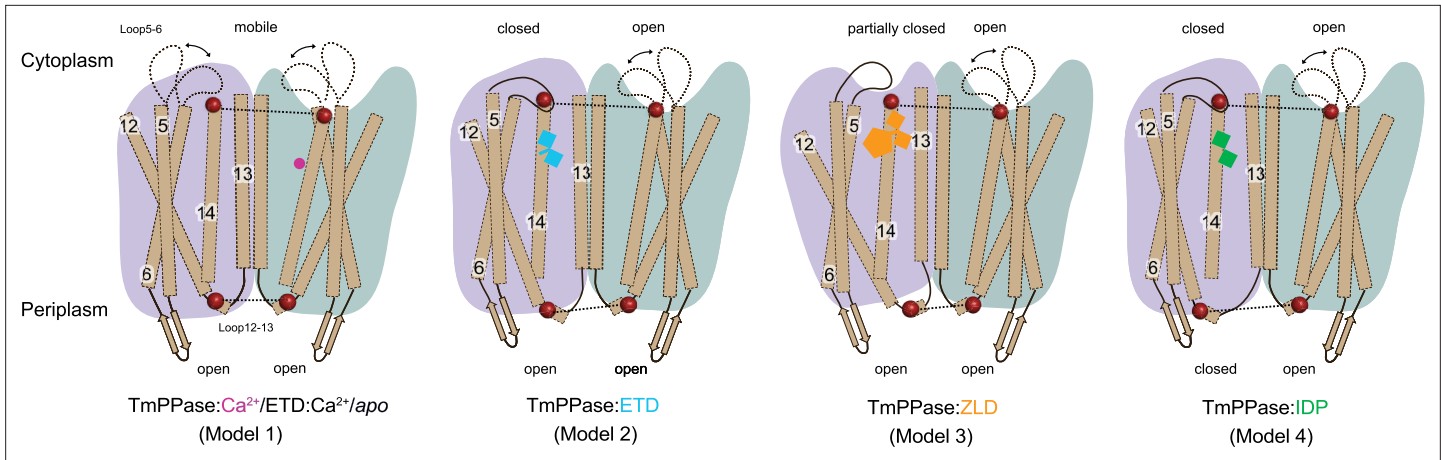

**Figure 6.** Models based on double electron-electron resonance (DEER) distance distributions for TmPPase S525R1 and C599R1. Four DEER models showing major conformational ensembles of TmPPase in solution. Two monomers are coloured purple and green, respectively. All transmembrane helices (TMHs) are shown in brown; mobile loop5–6 is indicated by a black dashed line, while fixed loop5–6 and loop12–13 are indicated by a solid black line. The labelling sites are represented by maroon spheres. $Ca^{2+}$ is shown as a magenta circle; imidodiphosphate (IDP) is shown as purple squares; etidronate (ETD) as cyan squares connected by a cyan stick; zoledronate (ZLD) as an orange pentagon.

This suggests that conformational differences on the cytoplasmic side between the DEER data and crystal structures are not significantly manifested at the exit channel.

In the presence of IDP, however, we observed a longest distance distribution (~4.0 nm), consistent with the predicted distance from the hybrid asymmetric TmPPase:IDP(A)_Ca(B) (*Figure 4A*) (closed-open), but neither the open-open nor closed-closed states. The ETD distance is intermediate, at ~3.9 nm (*Figure 4D*), suggesting that a complete change to the closed conformational state on the periplasmic side does not occur, consistent with absence of $Na^+$ in the exit channel. In contrast, with ZLD bound, the DEER distance distribution is the shortest (3.4 nm) (*Figure 4D*) and significantly deviates from the predicted distance for TmPPase:ZLD structure. This discrepancy may arise because, in solution, while ZLD can enter the active site, its bulky heteroaryl group, which orients towards TMH12 (*Figure 3D*), prevents the full downward movement of this helix. This structural restriction results in a shorter DEER distance distribution. For PAM and ALE, the DEER distance distributions are even longer than those observed for IDP, closely matching the TmPPase:IDP structure. Since we do not have structures for their complexes with TmPPase, their orientation in the active site remains unknown.

## Sodium ion pumping in TmPPase

Taken together, the X-ray crystallography and solution-state DEER data were used to propose a schematic for conformational transitions upon the addition of different compounds (*Figure 6*). Model 1 represents an asymmetric state at the cytoplasmic side under *apo*, +Ca, and +Ca/ETD conditions. Loops5–6 are highly flexible, consistent with the broad distribution observed in DEER data for C211R1 and the missing electron densities in crystal structures. The periplasmic side remains in the 'open' state, with helices 12 and 16 'up', consistent with the solved structures (*Kellosalo et al., 2012*). Model 2 describes the structural effects of ETD binding. C599R1, located at TMH14, reports a 'closed-open' state, with ligand binding to just one active site. However, there is no complete conformational change on the periplasmic side; the conformation is 'open-open'. Model 3, with ZLD, features the bulky heteroaryl group of ETD pulling the TMH 12 away at the cytoplasmic side, further affecting its conformation at the periplasmic side in an 'open-open' state. Model 4, with IDP, induces a 'closed-open' state at the cytoplasmic side, with ligand binding to just one active site, and also drives a full downward movement of TMH12 in one monomer. This conformational shift results in an asymmetric conformation at the exit channel, while the other monomer remains open, consistent with the 'closed-open' hybrid structure TmPPase:IDP(A)_Ca(B) (*Figure 4A*).

In a previous study (*Strauss et al., 2024*), we found that a single turnover event of $Na^+$ pumping only occurs in the presence of IDP. In our current Nanion SURFE$^2$R experiment, we did not observe $Na^+$ pumping (*Figure 5A*) upon the addition of ETD and ZLD, consistent with ETD- and ZLD-bound

structures where no $Na^+$ was observed at the ion gate. These data are consistent with the models presented above (*Figure 6*): IDP generates an asymmetric conformation in both the active site and in the exit channel, which occurs through the motion of TMH12. TMH5, TMH13, and TMH10 are key parts of intra-subunit communication between the two monomers (*Strauss et al., 2024*). (Loop12–13, where S525R1 is located, can be used to monitor the motion of TMH12.) However, neither ETD nor ZLD generate any Nanion SURFE$^2$R signal; the structures with these ligands do not reveal $Na^+$ at the ionic gate. This is completely consistent with the C599R1 DEER distance distributions (see Results), indicating that the cytoplasmic side (C599R1) is consistent with the 'closed-open' asymmetric conformation but that this has not propagated fully to the periplasmic side (S525R1), which is in the symmetric 'open-open' conformation, consistent with the Bhattacharyya coefficients, and which does not bind $Na^+$ at the ion gate. Consequently, the distance of 4.0 nm at S525R1, as observed in the IDP-bound sample, likely represents the minimal structural arrangement distance required for $Na^+$ pumping.

The DEER data thus provide a convincing structural explanation for why TmPPase is unable to pump $Na^+$ upon the addition of ETD or ZLD. In summary, EPR experiments in solution, coupled with new structures of inhibited forms of TmPPase, provide evidence supporting symmetry-breaking across the membrane, consistent with half-of-the-sites-reactivity (*Strauss et al., 2024*). In future studies, we will use time-resolved DEER to explore the order of conformational changes and how substrate addition is correlated with the release of product phosphate and ion pumping.

Note: During the revision of this manuscript, *Anashkin et al., 2025*, published a stopped-flow analysis demonstrating that the proton pumping in mPPase from *D. hafniense* only occurs in the presence of $PP_i$, as measured by fluorescence changes in the pH-sensitive dye pyranine. In comparison, our Nanion SURFE$^2$R can also detect signals induced by partial ion pumping or charged amino acid rearrangement, rather than solely ion pumping. The half reduction in signal in the presence of IDP may be due to $Na^+$ being translocated to the ion gate and locked there without further release, consistent with the TmPPase:IDP structure and our DEER data. The weak signals observed in the presence of ETD or ZLD are likely due to charged amino acid rearrangements induced by their binding.

# Materials and methods

## Key resources table

| Reagent type (species) or resource | Designation | Source or reference | Identifiers | Additional information |
|---|---|---|---|---|
| Gene (*Thermotoga maritima*) | hppA | GeneBank | AAD35267.1 | |
| Strain, strain background (*Saccharomyces cerevisiae*) | BJ1991 | ATCC | 208275 | |
| Recombinant DNA reagent | TmPPase-pRS1024 | Serrano R, Kielland-Brandt MC, Fink GR. Yeast plasma membrane ATPase is essential for growth and has homology with (Na$^{++}$ K$^+$), K$^+$- and Ca$^{2+}$-ATPases. *Serrano et al., 1986* Feb 20–26;319 (6055):689–93. doi: https://doi.org/10.1038/319689a0. PMID:3005867. | | Gift from Professor A Serrano (University of Sevilla) |
| Chemical compound, drug | Imidodiphosphate sodium salt (IDP) | Sigma-Aldrich | I0631-1G | |
| Chemical compound, drug | Disodium etidronate hydrate | TCI | D4159 | |
| Chemical compound, drug | Pamidronate disodium salt hydrate | TCI | D3921 | |
| Chemical compound, drug | Alendronate sodium trihydrate | TCI | A2456 | |
| Chemical compound, drug | Neridronate | Sigma-Aldrich | N6037 | |
| Chemical compound, drug | Ibandronate sodium | TCI | S0877 | |
| Chemical compound, drug | Zoledronic acid monohydrate | TCI | Z0031 | |

*Continued on next page*

Continued

| Reagent type (species) or resource | Designation | Source or reference | Identifiers | Additional information |
|---|---|---|---|---|
| Chemical compound, drug | Monosodium risedronate hemipentahydrate | TCI | M2289 | |

## Protein expression and purification

TmPPase expression and purification have been described previously (*Kellosalo et al., 2011*; *López-Marqués et al., 2005*). Briefly, the pRS1024 plasmid containing His-tagged TmPPase was freshly transformed into *Saccharomyces cerevisiae* strain BJ1991. The cells were cultured in 250 ml of selective synthetic complete drop-out (SCD) media overnight before being added to 740 ml of 1.5× YP media with 2% glucose. The cells were then cultured for 8 hr at 30°C, collected by centrifugation (4000 rpm, 10 min) and lysed at 4°C using a bead beater with 0.2 mm glass beads. The membrane fraction was collected by ultracentrifugation (100,000×$g$, 45 min) and the pellets were resuspended in buffer containing 50 mM MES-NaOH pH 6.5, 20% (vol/vol) glycerol, 50 mM KCl, 5.2 mM MgCl$_2$, 1.33 mM dithiothreitol (DTT), 2 µg/ml (wt/vol) pepstatin-A (Sigma), and 0.334 mM PMSF (Sigma). The membranes were solubilised in solubilisation buffer (50 mM MES-NaOH pH 6.5, 20% [vol/vol] glycerol, 5.33% [wt/vol] n-dodecyl-β-D-maltopyranoside [DDM] [Anatrace]) using the 'hot-solve' method (*López-Marqués et al., 2005*) at 75°C for 1.5 hr. After centrifugation to remove denatured proteins, KCl (to a final concentration of 0.3 M) and 2 ml of Ni-NTA beads (QIAGEN) were added and incubated at 40°C for 1.5 hr, and then loaded into an Econo-Pac column (Bio-Rad). Then, the column was washed with two column volumes (CV) of washing buffer (50 mM MES-NaOH pH 6.5), 20% (vol/vol) glycerol, 50 mM KCl, 20 mM imidazole pH 6.5, 5 mM MgCl$_2$, 1 mM DTT, 2 mg/ml (wt/vol) pepstatin-A, 0.2 mM PMSF, and 0.05% DDM (Anatrace), and eluted with 2 CV of elution buffer (50 mM MES-NaOH pH 6.5, 3.5%) (vol/vol) glycerol, 50 mM KCl, 400 mM imidazole pH 6.5, 5 mM MgCl$_2$, 1 mM DTT, 2 mg/ml (wt/vol) pepstatin-A, 0.2 mM PMSF, and 0.5% octyl glucose neopentyl glycol (OGNPG, Anatrace).

## TmPPase activity assay

TmPPase activity and bisphosphonates inhibition assay were performed using the molybdenum blue reaction method in 96-well plate format as reported previously (*Vidilaseris et al., 2019a*). Before the assay, the enzyme was reactivated by adding to the mixture of 30 mg/ml of soy-bean lecithin (Sigma) in 20 mM Tris-HCl pH 8.0 with 4.5% DDM and incubated at 55°C for 15 min. The activity reaction was done in the reaction buffer (60 mM Tris-HCl pH 8.0, 5 mM MgCl$_2$, 100 mM KCl, and 10 mM NaCl) and started by adding 2 mM Na$_4$PP$_i$ at 71°C for 5 min.

## NEM modification assay

The NEM modification assay was performed as reported previously with slight modification (*Kellosalo et al., 2012*). Briefly, 0.4 mg/ml of the reactivated TmPPase was mixed with the modification buffer (20 mM MES-KOH pH 6.5, 0.05% DDM, 2.4 mM MgCl$_2$, 100 mM KCl, and 20 mM NaCl) and different inhibitors (2 mM CaCl$_2$, 0.5 mM IDP, and 0.5 mM of bisphosphonates) and incubated on ice for 30 min. Afterwards, 100 mM NEM (Thermo Scientific) was added and the mixture was further incubated for 10 min. The NEM-modification reactions were stopped by adding 2 mM DTT, and the residual activity of the enzyme was performed using the molybdenum blue reaction assay after removing excess inhibitors (*Vidilaseris et al., 2018*).

## Crystallisation and structure determination

For co-crystallisation with bisphosphonates, the purified TmPPase was buffer-exchanged to the crystallisation buffer (50 mM MES-NaOH pH 6.5, 3.5% [vol/vol] glycerol, 50 mM KCl, 5 mM MgCl$_2$, 2 mM DTT, and 0.5% OGNPG) on a Micro Bio-Spin 6 column (Bio-Rad) and then diluted to a concentration of 10 mg/ml. Prior to crystallisation, 1 mM bisphosphonates was added to the protein solution, incubated at room temperature for 30 min, and centrifuged for 20 min (16,000×$g$, 4°C). Crystallisation trials were done using a Mosquito robot (SPT Labtech) by sitting drop vapour-diffusion method using MemGold screen (Molecular Dimensions) in MRC two-well crystallisation plates (Swissci), and the drops were monitored at 22°C using the minstrel DT UV imaging system (Formulatrix). Crystal hits appeared on the MemGold screen under different conditions. Harvestable crystals appeared within several days

and were frozen directly from the mother liquor. For the TmPPase co-crystallised with etidronate, the best diffracting crystal was observed from a solution containing 0.2 M CaCl$_2$, 0.1 M HEPES pH 7.0, and 33% PEG400, while for TmPPase co-crystallised with zoledronate, the best diffracting crystal was observed from a solution containing 0.1 M MES pH 6.5, 0.1 M NaCl, 33% PEG400, and 4% ethylene glycol.

X-ray diffraction data were collected at Diamond Light Source (DLS) (UK) on the I03 (TmPPase:ETD) and I04-1 beamline (TmPPase:ZLD) at 100 K on a Pilatus 6 M detector. The data were merged and scaled using X-ray Detector Software (XDS) (*Kabsch, 2010*), and the structure was solved by molecular replacement with Phaser (*McCoy et al., 2007*) using the resting state (4AV3) (*Kellosalo et al., 2012*) and IDP-bound (5LZQ) state (*Li et al., 2016*) of TmPPase structure as the search model for TmPPase:Etidronate and TmPPase:Zoledronate, respectively. The structures were built and refined using phenix.refine (*Adams et al., 2010*) and Coot (*Emsley et al., 2010*). X-ray data and refinement statistics are listed in *Supplementary file 1*.

## EPR spectroscopy
### Sample preparation for EPR spectroscopy
For EPR spectroscopy measurements, we utilised a nearly Cys-less construct, retaining only endogenous cysteine C183 due to its buried location and functional importance. Residue S525, located in the periplasmic loop12–13 of the TmPPase exit channel, was mutated to cysteine and covalently modified with a methanethiosulphonate thiol-specific spin label (MTSSL) to introduce a paramagnetic centre (*Hubbell et al., 1998*; *Jeschke, 2012*) (the labelled protein is referred to as S525R1). At the cytoplasmic side of the membrane interface, we constructed the TmPPase T211C variant, which is located in loop5–6 and above the active site (the MTSSL labelled mutant is referred to as T211R1). We also spin-labelled an endogenous cysteine residue, C599 (after mutating back the S599 to cysteine) on the cytoplasmic transmembrane helix 14 (the labelled protein is referred to as C599R1).

The S525C, 599C, and T211C proteins were expressed as outlined above. The frozen cell pellets were lysed using cryo-milling (Retsch model MM400). 1 mM TCEP was used to replace DTT in the purification steps preceding spin labelling, and the remaining purification was carried out as above. Each protein was spin-labelled with MTSSL while immobilised to the Ni-NTA resin (or mixed following Cys mutant elution) as previously described (*Lane et al., 2024*; *Lane et al., 2022*). Briefly, for MTSSL labelling, MTSSL was added in spin-label buffer (20 mM MOPS-NaOH, 5 mM MgCl$_2$, 50 mM KCl, 3.5% glycerol, 0.03% DDM at pH 7.5) at 10-fold molar protein excess and incubated for 2 hr at room temperature. For C599, 10 mM CaCl$_2$ was added in the buffer to increase the accessibility of the site for spin-labelling (i.e. to induce partial opening). Spin-labelled protein was eluted from the Ni-NTA resin column, concentrated and subsequently purified using size-exclusion chromatography using Superose 6 increase 10/300 GL (GE Healthcare) and equilibrated in 20 mM MES-NaOH, pH 6.5, 5 mM MgCl$_2$, 50 mM KCl, 3.5% glycerol, 0.05% DDM. The eluted purified protein fractions were concentrated, buffer-exchanged with buffer prepared in D$_2$O, and split into aliquots for incubation with a final concentration of 2 mM of all inhibitors or 10 mM CaCl$_2$ (30 min, room temperature). The protein activity was tested as described above, and the protein samples were tested for spin labelling by CW-EPR spectroscopy, and then 40% ethylene glycol-$d_6$ was added to each sample before flash-freezing for DEER measurement.

### CW-EPR spectroscopy
CW-EPR experiments were performed on a Bruker Magnettech ESR5000 X-band spectrometer (9.4 GHz). The spin-labelled sample was loaded into a 3 mm (o.d.) quartz EPR tube before the addition of ethylene glycol-$d_6$. The samples were measured at room temperature (298 K), as TmPPase is more thermally stable than most membrane proteins. The measurements were performed in a magnetic field range of 330–345 mT, with a 60 s sweep time, 0.1 mT modulation amplitude, 100 kHz modulation frequency, and 10 mW (10 dB) microwave power.

## DEER (or PELDOR) spectroscopy
### DEER distance measurements and set-up
EPR recordings were collected as previously described (*Milov et al., 1984*) using a Bruker ELEXSYS E580 spectrometer operating at Q-band (34 GHz) frequency, equipped with a QT-II resonator in a

cryogen-free variable temperature cryostat (Cryogenic Ltd.) with a temperature range of 2–300 K. In brief, spin-labelled protein samples were prepared in 3 mm outer diameter quartz tubes, and data was recorded at 50 K. The detection pulse sequence used was a refocused Hahn echo: $\pi/2 - \tau 1 - \pi - \tau 1 - \tau 2 - \pi - \tau 2 -$ echo, with $\pi/2$ and $\pi$ observer pulse lengths of 16 and 32 ns, and $\pi$ inversion pulse lengths of 16–20 ns, $\tau 1$ of 380 ns, and $\tau 2$ of 2000–5000 ns, depending on construct. Unless otherwise stated, the magnetic field and microwave frequency were adjusted for the maximum of the nitroxide spectrum to coincide with the pump pulse position, while the observer pulse was placed at either 65 MHz (for T211R1 and S525R1 measurements) or 80 MHz (C599R1 measurements) frequency offset. Measurements were recorded using either a 150 W (for T211R1 and S525R1 measurements) or a 300 W (for C599R1 measurements) travelling wave tube (TWT; Applied Systems Engineering). All pulses were generated using an integrated Bruker SpinJet AWG, and measurements were recorded using a 16-step phase cycle on the detection pulses to remove unwanted echo crossings (*Tait and Stoll, 2016*). Finally, electron-spin echo envelope modulation arising from electron-nuclear coupling to deuterium was suppressed using an eight-step tau-averaging cycle (*Keller et al., 2016*), with a time increment of 16 ns.

For the measurements of S525R1 TmPPase (excluding the +PAM and +ALE conditions), a 4 ns dipolar increment was used to yield the DEER trace. Owing to significant excitation bandwidth overlap between observer and pump pulses at low-frequency offset (–65 MHz), the presence of a '2+1' artefact exacerbated data treatment. Therefore, traces were recorded using a $\tau 2$ of 5000 ns (and truncated to 4000 ns for data processing [see DEER data analysis and processing section below]), 16 shots-per-point, 647 points, and a shot repetition time (SRT) of 3060 µs. Scans were recorded until a sufficient signal-to-noise ratio was obtained, typically with datasets averaged overnight. For the measurements of S525R1 TmPPase +PAM and +ALE, a 300 W TWT (Applied Systems Engineering) was used, operating at Q-band frequency. Traces were recorded with a 12 ns dipolar increment using a $\tau 2$ of 4000 ns, 10 shots-per-point, 348 points, and SRT of 2000 µs.

For the measurements of C599R1 TmPPase, a 12 ns dipolar increment was used, and traces were recorded using a $\tau 2$ of 5000 ns, 10 shots-per-point, 432 points, and an SRT of 2000 µs. Scans were recorded until a sufficient signal-to-noise ratio was obtained, typically with datasets averaged overnight. For all measurements of T211R1 (excluding the apo and +Ca/ETD measurements), a 4 ns dipolar increment was used, and traces were recorded using a $\tau 2$ of 2000 ns, 32 shots-per-point, 432 points, and an SRT of 3000 µs. Finally, for the apo and +Ca/ETD measurements, a 4 ns dipolar increment was used, and traces were recorded using a $\tau 2$ of 4000 ns, 16 shots-per-point, 525 points, and an SRT of 3000 µs.

## DEER data analysis and processing

Distance distributions were determined from the time traces using various methodologies as best practices to get reliable results and to ensure self-consistency (*Schiemann et al., 2021*). In the present work, we used two different programs, DeerAnalysis2022 (*Schiemann et al., 2021*; *Russell et al., 2022*) and ComparativeDeerAnalyzer2.0 (*Schiemann et al., 2021*), with results in the main text corresponding to the DeerAnalysis2022 processing. The 525R1 and T211R1 data recorded with a low-frequency offset (65 MHz) yielded strong '2+1' artefacts, owing to overlapping pulse excitation profiles. To address this, all 525R1 datasets, and the apo and +Ca/ETD measurements for T211R1, were truncated or recorded to 4000 ns (*Teucher and Bordignon, 2018*), respectively, and then phase and background-corrected using the '!' automated adjustment. The background-corrected traces were then transformed from the time domain to the distance domain using Tikhonov regularisation (*Chiang et al., 2005*), and the quality of the fit was assessed based on the L-curve criterion and the shape of the Pake pattern. The resultant background correction was then validated using a module for Tikhonov validation implemented in DeerAnalysis2022. The validation was carried out after initial Tikhonov regularisation, varying the background start time from 5% to 80% of the respective time windows of the cut data for 16 trials. From this, the raw data were re-loaded and processed (Tikhonov regularisation) with the cutoff and background start time as established from the first round of validation. This is the starting point for a full validation, where the background start time was again varied from 5% to 80% of the time window for 16 trials, as well as some added 'white noise' with a level of 1.50 for 50 trials. The resulting validation trials were pruned and yielded the distance distribution and confidence interval. The ComparativeDeerAnalyzer2.0 (CDA) was used to automate data processing

and reduce operator bias. The corresponding output data for S525R1 and C599R1 TmPPase are shown in *Figure 4—figure supplement 3*.

Methods for B coefficients calculation (*Bhattacharyya, 1946*): Bhattacharyya coefficients were used as similarity metric between experimental apo-state distribution and the in silico distribution predicted from the asymmetric hybrid structure. Following *Equation 1*:

$$BC = \sum_{n \in N} \sqrt{P(n) \cdot Q(n)}$$

(1)

where P(n) and Q(n) are the normalised probability distributions (i.e. to convert from probability density distributions to probability distributions (*Equation 2*)) on the same domain N.

$$P(n) = \frac{P(n)}{\sum_{n \in N} P(n)}$$

(2)

## In silico spin labelling and modelling

MttslWizard (*Hagelueken et al., 2012*) and ChiLife (*Tessmer and Stoll, 2023*) were used to predict in silico distance distributions for the T211R1, and S525R1 and C599R1 labelling sites of TmPPase, respectively. The coordinates of the respective X-ray structures were TmPPase:Ca PDB 4AV3, TmPPase:Ca:ETD PDB 9G8K, TmPPase:ZLD PDB 9G8J, and TmPPase:IDP PDB 5LZQ for the different conditions were uploaded to the online MTSSL Suite server, labelled at T211 sites (both monomers A and B) with R1 and the rotamer cloud was generated using the 'tight' labelling mode (i.e. zero steric clashes allowed). The PDB structures (including the asymmetric hybrid TmPPase_Ca:TmPPase_IDP structure) were also loaded into ChiLife and R1 side chains were introduced, individually, at S525 and C599 sites. For consistency with MTSSLWizard predictions, the accessible volume approach to calculate rotamer clouds was used.

## Electrometric measurement

For the Nanion SURFE$^2$R experiment, purified TmPPase was reconstituted into liposomes as previously described with some modifications (*Li et al., 2016*). Briefly, the purified protein was buffer-exchanged into a reconstitution buffer (50 mM MOPS-KOH pH 7.2, 50 mM KCl, 5 mM $MgCl_2$, and 2 mM DTT) to remove $Na^+$ and glycerol and then diluted to 50 mg/ml concentration. 15 ml of liposome solution (120 mg/ml soy-bean lecithin in 50 mM MOPS-KOH pH 7.2) was mixed with 1 ml of diluted protein sample. SM-2 Bio-beads were added in increments to a final concentration of 0.25 mg/ml and then placed into a mixer at 4°C for 6 hr to ensure beads stayed in suspension. The proteoliposomes were collected and frozen at –80°C in aliquots. To ensure that the reconstituted protein was still active, the hydrolytic activity was performed using the molybdenum blue reaction assay (*Vidilaseris et al., 2018*).

Electrometric measurements were performed on a SURFE$^2$R N1 instrument from Nanion Technology (Munich, Germany). The gold sensors were prepared based on the 'SURFE$^2$R N1 protocol', including thiolation of the sensor surface and assembly of the lipid layer using sensor prep A2 and B solutions. 15 ml of sonicated proteoliposomes, followed by 50 ml of the rinsing buffer (50 mM MOPS-KOH pH 7.2, 50 mM NaCl, 5 mM $MgCl_2$), were applied directly to the sensor surface. Sensors were centrifuged for 30 min at 2500×$g$ and incubated at 4°C for 3 hr. The sensors were mounted in the SURFE$^2$R N1 and rinsed once with 1 ml of rinsing buffer (50 mM MOPS-KOH, pH 7.2, 50 mM NaCl, 5 mM $MgCl_2$). Measurements were performed for 3 s by consecutively flowing non-activating buffer B (50 mM MOPS-KOH pH 7.2, 50 mM NaCl, 5 mM $MgCl_2$, 200 μM $K_2HPO_4$) and activating buffer A (50 mM MOPS-KOH, 50 mM NaCl, 5 mM $MgCl_2$) containing substrate (100 mM $K_4PP_i$) or inhibitors (50 mM IDP, 50 mM ETD, or 50 mM ZLD) across the sensor for 1 s each in a BAB sequence. Charge transport across the membrane is initiated by substrate or inhibitor in buffer A, which flows across the sensor between 1 and 2 s. The transport of positively charged ions during this period results in a positive electrical current, the signal output of the SURFE$^2$R N1 instrument. Between each measurement, the sensor was washed with 1 ml rinsing buffer and incubated for 60 s. The measurements were tested in triplicates.

## Acknowledgements

This work was supported by the Biotechnology and Biological Research Council (BBSRC) (BB/T006048/1) awarded to CP and AG Grants from the Research Council of Finland (No. 1322609 and 13364501) to AG, (No. 308105 and 1355187) to KV, and (No. 310297) to HX, also supported part of this work. The first author is funded by the China Scholarship Council (CSC) from the Ministry of Education of P.R. China. The authors thank Juho Kellosalo for fruitful discussions during the project. EPR measurements were performed at the national EPR facilities in Manchester and the BioEmPiRe Centre for Structural Biological EPR spectroscopy funded by BBSRC (BB/W019795/1) to CP. We thank DLS for access to beamline I03 and I04-1. The facilities and expertise of the HiLIFE Crystallization unit at the University of Helsinki, a member of FINStruct and Biocenter Finland, are gratefully acknowledged.

## Additional information

### Funding

| Funder | Grant reference number | Author |
|---|---|---|
| Biotechnology and Biological Sciences Research Council | BB/T006048/1 | Adrian Goldman<br>Christos Pliotas |
| Research Council of Finland | 1322609 | Adrian Goldman |
| Research Council of Finland | 13364501 | Adrian Goldman |
| Research Council of Finland | 308105 | Keni Vidilaseris |
| Research Council of Finland | 1355187 | Keni Vidilaseris |
| China Scholarship Council | | Jianing Liu |
| Biotechnology and Biological Sciences Research Council | BB/W019795/1 | Christos Pliotas |
| Research Council of Finland | 310297 | Henri Xhaard |

The funders had no role in study design, data collection and interpretation, or the decision to submit the work for publication.

### Author contributions

Jianing Liu, Formal analysis, Funding acquisition, Investigation, Visualization, Methodology, Writing – original draft, Writing – review and editing; Anokhi Shah, Investigation, Visualization, Methodology, Writing – original draft, Writing – review and editing; Xinyu Liu, Yue Ma, Formal analysis, Investigation, Visualization, Methodology, Writing – review and editing; Joshua L Wort, Validation, Investigation, Visualization, Methodology, Writing – review and editing; Katie Hardman, Adam Brookfield, Alice Bowen, Methodology; Niklas G Johansson, Resources, Methodology, Writing – review and editing; Orquidea Ribeiro, Investigation; Jari Yli-Kauhaluoma, Resources, Funding acquisition, Project administration, Writing – review and editing; Henri Xhaard, Conceptualization, Resources, Funding acquisition, Project administration, Writing – review and editing; Lars JC Jeuken, Methodology, Writing – review and editing; Adrian Goldman, Conceptualization, Resources, Supervision, Funding acquisition, Methodology, Project administration, Writing – review and editing; Christos Pliotas, Conceptualization, Resources, Formal analysis, Supervision, Funding acquisition, Validation, Investigation, Visualization, Methodology, Writing – original draft, Project administration, Writing – review and editing; Keni Vidilaseris, Conceptualization, Resources, Data curation, Formal analysis, Funding acquisition, Validation, Investigation, Visualization, Methodology, Writing – original draft, Project administration, Writing – review and editing

## Author ORCIDs

Jianing Liu (ID) http://orcid.org/0000-0003-0079-4712
Xinyu Liu (ID) https://orcid.org/0009-0008-6215-9101
Joshua L Wort (ID) http://orcid.org/0009-0001-9162-0429
Niklas G Johansson (ID) https://orcid.org/0000-0002-8226-4813
Jari Yli-Kauhaluoma (ID) https://orcid.org/0000-0003-0370-7653
Lars JC Jeuken (ID) https://orcid.org/0000-0001-7810-3964
Adrian Goldman (ID) https://orcid.org/0000-0001-8032-9700
Christos Pliotas (ID) https://orcid.org/0000-0002-4309-4858
Keni Vidilaseris (ID) https://orcid.org/0000-0002-6453-6600

Reviewer #1 (Public review): https://doi.org/10.7554/eLife.102288.3.sa1
Reviewer #3 (Public review): https://doi.org/10.7554/eLife.102288.3.sa2
Author response https://doi.org/10.7554/eLife.102288.3.sa3

## Additional files

### Supplementary files

Supplementary file 1. X-ray data collection and refinement statistics.

Supplementary file 2. Structural alignments between chains of different TmPPase structures.

MDAR checklist

### Data availability

The atomic coordinates and structure factors of the TmPPase:Etidronate and TmPPase:Zoledronate complex have been deposited in the Protein Data Bank, https://www.rcsb.org/ (PDB ID: 9G8K and 9G8J). The enzymatic activity and PELDOR/DEER datasets are available in the Dryad repository (DOI: https://doi.org/10.5061/dryad.m905qfvfj).

The following datasets were generated:

| Author(s) | Year | Dataset title | Dataset URL | Database and Identifier |
|---|---|---|---|---|
| Vidilaseris K, Liu J, Goldman A | 2025 | Structure of K+-dependent Na+-PPase from Thermotoga maritima in complex with Ca2+ and Etidronate | https://doi.org/10.2210/pdb9G8K/pdb | Worldwide Protein Data Bank, 10.2210/pdb9G8K/pdb |
| Vidilaseris K, Liu J, Goldman A | 2025 | Structure of K+-dependent Na+-PPase from Thermotoga maritima in complex with zoledronate | https://doi.org/10.2210/pdb9G8J/pdb | Worldwide Protein Data Bank, 10.2210/pdb9G8J/pdb |
| Liu J, Shah A, Liu X, Wort J, Ma Y, Hardman K, Johansson NG, Ribeiro O, Brookfield A, Bowen A, Yli-Kauhaluoma J, Xhaard H, Jeuken L, Goldman A, Pliotas C, Vidilaseris K | 2025 | Data from: Conformational dynamics and asymmetry in multimodal inhibition of membrane-bound pyrophosphatases | https://doi.org/10.5061/dryad.m905qfvfj | Dryad Digital Repository, 10.5061/dryad.m905qfvfj |

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
