## [Editor Report · eLife Assessment]

This **important** study uncovers the mechanism of inhibition of a membrane pyrophosphatase by non-hydrolyzable phosphonate substrate analogs. **Convincing** crystallography, EPR spectroscopy, and functional measurements support the presence of a distinct conformational equilibrium of TmPPase in solution, and further supports the notion of asymmetric inhibitor binding at the active site, while maintaining a symmetric conformation at the periplasmic interface.

---

## [Referee Report · Reviewer #1 (Public review)]

Summary:

This work examines the binding of several phosphonate compounds to a membrane-bound pyrophosphatase using several different approaches, including crystallography, electron paramagnetic resonance spectroscopy, and functional measurements of ion pumping and pyrophosphatase activity. The work synthesizes these different approaches into a model of inhibition by phosphonates in which the two subunits of the functional dimer interact differently with the phosphonate. This asymmetry in the two subunits of the dimer is consistent with past studies of this system.

Strengths:

This study integrates a variety of approaches, including structural biology, spectroscopic measurements of protein dynamics, and functional measurements. Overall, data analysis was thoughtful, with careful analysis of the substrate binding sites (for example calculation of POLDOR omit maps). This study agrees with previous studies that have detected functional asymmetry in the membrane PPase dimer.

---

## [Referee Report · Reviewer #3 (Public review)]

Summary:

Membrane-bound pyrophosphatases (mPPases) are homodimeric proteins that hydrolyze pyrophosphate and pump H+/Na+ across membranes. They are an attractive drug target against protist pathogens. Non-hydrolysable PPi analogue bisphosphonates such as risedronate (RSD) and pamidronate (PMD) serve as primary drugs currently used. Bisphosphonates have a P-C-P bond, with their central carbon can accommodate up to two substituents, allowing a large compound variability. Here authors solved two TmPPase structures in complex with the bisphosphonates etidronate (ETD) and zoledronate (ZLD) and monitored their conformational ensemble using DEER spectroscopy in solution. These results reveal the inhibition mechanism by these compounds, which is crucial for developing future small-molecule inhibitors.

Strengths:

Authors show that seven different bisphosphonates can inhibit TmPPase with IC50 values in the micromolar range. Branched aliphatic and aromatic modifications showed weaker inhibition. High-resolution structures for TmPPase with ETD (3.2 Å) and ZLD (3.3 Å) are determined. These structures reveal the binding mode and shed light on the inhibition mechanism. The nature of modification on the bisphosphonate alters the conformation of the binding pocket. The conformational heterogeneity is further investigated using EPR/DEER spectroscopy under several conditions. Altogether, this provides convincing evidence for a distinct conformational equilibrium of TmPPase in solution and further supports the notion of asymmetric inhibitor binding at the active site, while maintaining a symmetric conformation at the periplasmic interface.

---

## [Author Response]

The following is the authors’ response to the original reviews

**Public Reviews:**

**Reviewer #1 (Public review):**
Summary:This work examines the binding of several phosphonate compounds to a membrane-bound pyrophosphatase using several different approaches, including crystallography, electron paramagnetic resonance spectroscopy, and functional measurements of ion pumping and pyrophosphatase activity. The work attempts to synthesize these different approaches into a model of inhibition by phosphonates in which the two subunits of the functional dimer interact differently with the phosphonate.Strengths:This study integrates a variety of approaches, including structural biology, spectroscopic measurements of protein dynamics, and functional measurements. Overall, data analysis was thoughtful, with careful analysis of the substrate binding sites (for example calculation of POLDOR omit maps).Weaknesses:Unfortunately, the protein did not crystallize with the more potent phosphonate inhibitors. Instead, structures were solved with two compounds with weak inhibitory constants >200 micromolar, which limits the molecular insight into compounds that could possibly be developed into small molecule inhibitors. Likewise, the authors choose to focus the spectroscopy experiments on these weaker binders, missing an opportunity to provide insight into the interaction between more potent binders and the protein.

We acknowledge the reviewer concern regarding the choice of weaker inhibitors. We attempted cocrystallization with all available inhibitors, including those with higher potency. However, despite numerous efforts, these potent inhibitors yielded low-resolution crystals, making them unsuitable for detailed structural analysis. Therefore, we chose to focus on the weaker binders, as we were able to obtain high-quality crystal structures for these compounds. This allowed us to perform DEER spectroscopy and monitor conformational TmPPase state ensembles in solution with the added advantage of accurately analysing the data against structural models derived from X-ray crystallography. Using these weaker inhibitors enabled a more precise interpretation of the DEER data, thus providing reliable insights into the conformational dynamics and inhibition mechanism. As suggested by the reviewer, in the revised version, we add new DEER experiments, conditions and analysis on two of the more potent inhibitors (alendronate and pamidronate) to provide additional insight into their interactions. Furthermore, we also implemented additional DEER data on the cytoplasmic side of TmPPase; at a new site we identified (with the advantage of being an endogenous cysteine residue) and spin labelled (C599R1), given the DEER data for the previous T211R1cytoplasmic site were difficult to interpret owing to the highly dynamic nature of this region. The new pair C599R1 yielded high-quality DEER traces and indicated more clearly than T211R1, distance distributions consistent with asymmetry across the sampled conditions. Again, as suggested by the reviewer, alendronate and pamidronate DEER measurements were also recorded for this site (cytoplasmic side; C599R1) as well as the periplasmic side (525R1).

In general, the manuscript falls short of providing any major new insight into membrane-bound pyrophosphatases, which are a very well-studied system. Subtle changes in the structures and ensemble distance distributions suggest that the molecular conformations might change a little bit under different conditions, but this isn't a very surprising outcome. It's not clear whether these changes are functionally important, or just part of the normal experimental/protein ensemble variation.

We respectfully disagree with the reviewer. The scale of motions particularly seen in solution (and now on a new reliable spin pair (C599R1) located on the cytoplasmic side) correspond to those seen in the full panoply of crystal structures of mPPases. Some proteins undergo very large conformational changes during catalysis – such as the rotary ATPase. This one does not, meaning that the precise motions we describe here are relevant and observed in solution for the first time. Conformational changes in the ensemble, whether large or small, represent essential protein motions which underlie key mPPase catalytic function. These dynamic transitions are extremely challenging to monitor, especially in so many conditions and our DEER spectroscopy data demonstrate the sensitivity and resolution necessary to monitor these subtle changes in equilibria, even if these are only a few Angstroms. For several of the conditions we investigated by DEER in solution, corresponding X-ray structures have been solved, with the derived distances agreeing well with the DEER distributions. This further validates the biological relevance of the structures, and reveals the complete conformational ensemble, intractable using other current approaches. Indeed, some conformational states were previously seen using serial time-resolved X-ray static structures and were consistent with asymmetry.

The ZLD-bound crystal structure doesn't predict the DEER distances, and the conformation of Na+ binding site sidechains in the ZLD structure doesn't predict whether sodium currents occur. This might suggest that the ZLD structure captures a conformation that does not recapitulate what is happening in solution/ a membrane.

We agree with the reviewer that the ZLD-bound crystal structure does not predict the DEER distances. However, we believe this discrepancy arises from the steric bulkiness of ZLD inhibitor, which prevents the closure of the hydrolytic centre. Additionally, the absence of Na+ at the ion gate in the ZLD-bound structure suggests that Na+ transport does not occur, a conclusion further supported by our electrometric measurements. We agree with the reviewer; distances observed in the DEER experiments might represent a potential new conformation in solution, not captured by the static X-ray structure, thereby offering new insights into the dynamic nature of the protein under physiological conditions. This serves to emphasize the complementarity of the DEER approach to Xray crystallography and redoubles the importance of using both techniques. Finally, the static X-ray structures have not captured the asymmetric conformations that must exist to explain half-of-thesites reactivity, where DEER yields distance distributions, across all 16 cases tested here (two mutants with eight conditions each), that are consistent with asymmetry.

**Reviewer #2 (Public review):**
Summary:Crystallographic analysis revealed the asymmetric conformation of the dimer in the inhibitor-bound state. Based on this result, which is consistent with previous time-resolved analysis, authors verified the dynamics and distance between spin introduced label by DEER spectroscopy in solution and predicted possible patterns of asymmetric dimer.Strengths:Crystal structures with inhibitor bound provide detailed coordination in the binding pocket thus useful information for the mPPase field and maybe for drug development.Weaknesses:The distance information measured by DEER is advantageous for verifying the dynamics and structure of membrane protein in solution. However, regarding T211 data, which, as the authors themselves stated, lacks measurement precision, it is unclear for readers how confident one can judge the conclusion leading from these data for the cytoplasmic side.

We thank the reviewer for acknowledging the advantageous use of the DEER methodology for identifying dynamic states of membrane proteins in solution. In our original manuscript, we used two sites in our analysis: S525 (periplasm) and T211 (cytoplasm), in which S525R1 yielded highquality DEER data, while T211R1 yielded weak (or no) visual oscillations, leading to broad distributions for the several conditions tested. In the revised manuscript, we now added a third site at the cytoplasmic side (C599R1 located at TMH14), which yielded high-quality DEER data and comparable to S525R1. Both C599R1 and C525R1 spin pairs generated distance distributions for all 16 conditions (two mutants of eight conditions each) that were described well by the solution-state ensemble adopting a predominantly asymmetric conformation.

Furthermore, we have tailored our interpretation of the T211R1 DEER data, and refrain from using the data to draw conclusions about the TmPPase conformational ensemble in the presence of different inhibitors. However, we still opted to include the T211R1 data in the SI because they confirm an important structural feature of mPPase in solution conditions; the intrinsically dynamic behaviour of the loop5-6 where T211 is located. This observation in solution is also consistent with our previous (Kellosalo et al., Science, 2012; Li et al., Nat. Commun, 2016; Vidilaseris et al., Sci. Adv., 2019; Strauss et al., EMBO Rep., 2024) and current X-ray crystallography data. To reiterate, we excluded T211R1 from any analysis relating to mPPase asymmetry and our conclusions were entirely based on the S525R1 and new C599R1 DEER data, which allowed us to monitor both sides on the membrane.

The distance information for the luminal site, which the authors claim is more accurate, does not indicate either the possibility or the basis for why it is the ensemble of two components and not simply a structure with a shorter distance than the crystal structure.

We thank the reviewer for pointing out this possibility and alternative interpretation of our DEER data. We now provide further analysis to show that our DEER data from both membrane sides reporters are highly consistent with (although they cannot completely exclude) asymmetry and rephrase to be inclusive of other possibilities. Importantly, this additional possibility does not affect the current interpretation of the data in our manuscript. Furthermore, we have removed Fig. 6 from the manuscript, and we now include a direct comparison of the in silico predicted distribution coming from the asymmetric hybrid structure with the 8 conditions tested, for both mutants (i.e. S525R1 and C599R1).

**Reviewer #3 (Public review):**
Summary:Membrane-bound pyrophosphatases (mPPases) are homodimeric proteins that hydrolyze pyrophosphate and pump H+/Na+ across membranes. They are attractive drug targets against protist pathogens. Non-hydrolysable PPi analogue bisphosphonates such as risedronate (RSD) and pamidronate (PMD) serve as primary drugs currently used. Bisphosphonates have a P-C-P bond, with its central carbon can accommodate up to two substituents, allowing a large compound variability. Here the authors solved two TmPPase structures in complex with the bisphosphonates etidronate (ETD) and zoledronate (ZLD) and monitored their conformational ensemble using DEER spectroscopy in solution. These results reveal the inhibition mechanism of these compounds, which is crucial for developing future small molecule inhibitors.Strengths:The authors show that seven different bisphosphonates can inhibit TmPPase with IC50 values in the micromolar range. Branched aliphatic and aromatic modifications showed weaker inhibition.High-resolution structures for TmPPase with ETD (3.2 Å) and ZLD (3.3 Å) are determined. These structures reveal the binding mode and shed light on the inhibition mechanism. The nature of modification on the bisphosphonate alters the conformation of the binding pocket.The conformational heterogeneity is further investigated using DEER spectroscopy under several conditions.Weaknesses:The authors observed asymmetry in the TmPPase-ELD structure above the hydrolytic center. The structural asymmetry arises due to differences in the orientation of ETD within each monomer at the active site. As a result, loop5-6 of the two monomers is oriented differently, resulting in the observed asymmetry. The authors attempt to further establish this asymmetry using DEER spectroscopy experiments. However, the (over)interpretation of these data leads to more confusion than any further understanding. DEER data suggest that the asymmetry observed in the TmPPase-ELD structure in this region might be funneled from the broad conformational space under the crystallization conditions.

We respectfully disagree with the reviewer. The asymmetry was previously established using serial time crystallography (Strauss et al., EMBO Rep, 2024) and biochemical assays (e.g. Malinen et al., Prot. Sci., 2022; Artukka et al., Biochem J, 2018; Luoto et al., PNAS, 2013) and partially seen in one static structure (Vidilaseris et al., Sci Adv 2019). DEER data here also show that the previously proposed asymmetry is also present (and this presence of asymmetry is consistent across all DEER data) within the TmPPase conformational ensemble in solution conditions. Although we cannot rule out the possibility that the TmPPase monomers adopt a metastable intermediate state, in such a case we would expect the distance changes reported by DEER to be symmetric across both membrane sides. However, we observe a symmetry breaking between the cytoplasmic and periplasmic TmPPase sites. Indeed, DEER data yield distance distributions similar to that of the hybrid asymmetric structure under all: apo, +Ca, +Ca/ETD, +ETD, +ZLD, +IDP, +PAM, +ALE conditions.

DEER data for position T211R1 at the enzyme entrance reveal a highly flexible conformation of loop56 (and do not provide any direct evidence for asymmetry, Figure EV8).

Please see relevant response above. We acknowledge that T211 is indeed situated on a highly dynamic loop, which is important for gating and our DEER data confirm the high flexibility of this protein region. Given we have not observed dipolar oscillations, leading to broad distributions, we have stated in the original manuscript that we will not establish the presence of any asymmetry in solution on the basis of T211, rather relying on the S525R1 and the new C599R1 sites, for which we have acquired high-quality DEER data, as was also pointed out and has been commented on by all reviewers. We have provided data at the C599R1 position (same cytoplasmic side as 211 for which we have now limited our analysis to a minimum) which further provides evidence for asymmetry, including two new conditions.

Similarly, data for position S521R1 near the exit channel do not directly support the proposed asymmetry for ETD.

The reviewer appears to suggest that we hold the S525R1 DEER data as direct proof of asymmetry; this is combative on the grounds that to directly prove asymmetry would require time-resolved DEER measurements, far beyond the scope of this work. Rather, we have applied DEER measurements to explore whether asymmetry (observed previously via time-resolved X-ray crystallography) is also present (or indeed a possibility) in solution. All our S525R1 and C599R1 DEER data (recorded for eight conditions) are consistent with asymmetry (see also detailed response above).

Despite the high quality of the data, they reveal a very similar distance distribution. The reported changes in distances are very small (+/- 0.3 nm), which can be accommodated by a change of spin label rotamer distribution alone. Further, these spin labels are located on a flexible loop, thereby making it difficult to directly relate any distance changes to the global conformation

We thank the reviewer for recognising the high quality of our DEER data for the S525R1 site which we now complement with a new pair on the cytoplasmic facing membrane side (C599R1) with DEER data of comparable quality as for S525R1, where visual oscillations in the raw traces for both spin pairs, as in our case, reportedly lead to highly accurate and reliable distributions, able to separate (in fortuitous cases) helical movements of only a few Angstroms (Peter et al., Nature Comms 13:4396, 2022; Klose et al., Biophys J 120:4842-4858, 2021). The ability of DEER/PELDOR offering near Angstrom resolution was also previously demonstrated by the acquisition and solution of highresolution multi-subunit spin-labelled membrane protein structures (Pliotas at al., PNAS, 2012; Pliotas et al., Nat Struct Mol Biol, 2015; Pliotas, Methods Enzymol, 2017) as well as its ability in detecting small (and of similar to mPPase magnitude) conformational changes in different integral membrane protein systems (Kapsalis et al., Nature Comms, 2019; Kubatova et al., PNAS, 2023; Schmidt et al., JACS, 2024; Lane et al., Structure, 2024; Hett et al., JACS, 2021; Zhao et al., Nature, 2024), occurring under different conditions and/or stimuli in solution and/or lipid environment. The changes here are not below the detection sensitivity of DEER e.g. ~ 7 Angstroms between the two modal distance extremes (+Ca vs +IDP for S525R1), and with all other conditions showing intermediate changes.

We agree with the reviewer that these changes are relatively small, but they are expected for membrane ion pumps. Indeed, none of the mPPase structures show helical movements of greater than half a turn, and that only in helices 6 and 12. There appear to be larger-scale loop closing motions of the 5-6 loop that includes T211, due to the presence of E217 which binds to one of the Mg^2+^ ions that coordinate the leaving group phosphate. This is, inter alia, the reason that this loop is so flexible: it cannot order before substrate is bound.

The reviewer suggests that the subtle distance shifts detected arise only from changes of label rotamer distribution. However, the concerted nature of the modal distance shifts with respect to multiple different conditions at a single labelling site strongly suggests that preferential rotamer orientations are not the cause. Indeed, for so many spin labels to undergo an arbitrary shift that the modal distance of the entire distribution changes – and in the absence of any conformational change – appears improbable. Here we have the resolution to detect such subtle differences by DEER, given there are unambiguous shifts in our time domain data (i.e. the position of the minimum of the first dipolar oscillation) (Fig 4) and these are reflected in the modal distances in the distributions. We also refrain from performing any quantitative analysis and use qualitative trends in modal distance shifts only; all which support our proposed model of a symmetry breaking across the membrane face. To further belabour this point, we do not quantify the DEER data (for instance through parametric fitting) to extract populations of different conformational states and we appreciate that to do so would be highly prone to error; however we do (and can, we feel without over-interpretation) assert that the modal distances shift.

The interpretations listed below are not supported by the data presented:(1) 'In the presence of Ca2+, the distance distribution shifts towards shorter distances, suggesting that the two monomers come closer at the periplasmic side, and consistent with the predicted distances derived from the TmPPase:Ca structure.'Problem: This is a far-stretched interpretation of a tiny change, which is not reliable for the reasons described in the paragraph above.

While the authors overall agree with the reviewer assessment that ±0.3 nm is a small (not a minor) change, there are literature examples quantifying (or using for quantification) distribution peaks separated by similar Δr. (Kubatova et al., PNAS, 2023; Schmidt et al., JACS, 2024; Hett et al., JACS, 2021; Zhao et al., Nature, 2024). However, the time-domain data clearly indicate the position of the first minimum of the dipolar oscillation shifts to shorter dipolar evolution time. The sensitivity of the time-domain data to subtle changes in dipolar coupling frequency is significantly improved compared to the distance distributions.

Importantly, we have fitted Gaussians to the experimental distance distributions of 525R1 output by the Comparative Deer Analyzer 2.0 and observed a change in the distribution width in presence of Ca2+, implying the rotameric freedom of the spin label is restricted. However, the CW-EPR for 525R1 indicate that the rotational correlation time of the spin label is highly consistent between conditions (the spectra are almost identical); this cannot be explained simply by rotameric preference of the spin label (as asserted by the reviewer 3), as there is no (further) immobilisation observed from the CW-EPR of apo-state (Figure EV9) to that in presence of Ca2+. Furthermore, in the absence of conformational changes, it is reasonable to assume (and demonstrable from the CW-EPR data) that the rotamer cloud should not significantly change between conditions. However, Gaussian fits of the two extreme cases yielding the longest (i.e., in presence of IDP) and shortest (in presence of ZLD) modal distances for the 525R1 DEER data indicated significant (i.e., above the noise floor after Tikhonov validation) probability density for the IDP condition at 50 Å (P(r) = 0.18). This occurs at four standard deviations above the mean of the Guassian fit to the +ZLD condition, which by random chance should occur with <0.007% probability.

As in previous response, the method can detect changes of such magnitude which are not small, but physiologically relevant and expected for integral membrane proteins, such as mPPases. Indeed, even in equal (or more) complex systems such as heptameric mechanosensitive channel proteins DEER provided sub-Angstrom accuracy, when a spin labelled high resolution XRC structure was solved (Pliotas et al., PNAS, 2012; Pliotas et al., Nat Struct Mol Biol, 2015). Despite this being an ideal case where DEER accuracy was experimentally validated another high-resolution structural method on modified membrane protein and is not very common it demonstrates the power of the method, especially when strong oscillations are present in the raw DEER data (as here for mPPase S525R1, and C599R1), even when multiple distances are present, Angstrom resolution is achievable in such challenging protein classes.

(2) 'Based on the DEER data on the IDP-bound TmPPase, we observed significant deviations between the experimental and the in silico distances derived from the TmPPase:IDP X-ray structure for both cytoplasmic- (T211R1) and periplasmic-end (S525R1) sites (Figure 4D and Figure EV8D). This deviation could be explained by the dimer adopting an asymmetric conformation under the physiological conditions used for DEER, with one monomer in a closed state and the other in an open state.'Problem: The authors are trying to establish asymmetry using the DEER data. Unfortunately, no significant difference is observed (between simulation and experiment) for position 525 as the authors claim (Figure 4D bottom panel). The observed difference for position 112 must be accounted for by the flexibility and the data provide no direct evidence for any asymmetry.

Reviewer 3 is incorrect in suggesting that we are trying to prove asymmetry through the DEER data. That is a well-known fact in the literature (e.g. Vidilaseris et al, Sci Adv 2019) where we show (1) that the exit channel inhibitor ATC (i.e. close to S525R1) binds better in solution to the TmPPase:PPi complex than the TmPPase:PPi_2_ complex, and (2) that ATC binds in an asymmetric fashion to the TmPPase:IDP_2_ complex with just one ATC dimer on one of the exit channels. We merely use the DEER data to support this well-established fact.

However, because we agree that the DEER data in presence of IDP does not provide direct proof for asymmetry; particularly for the cytoplasmic facing mutant T211R1, we have refrained from interpreting T211R1 data beyond being a highly dynamic loop region (as evidenced by the broad distributions). As pointed out by the reviewer, the differences in distance distributions between conditions observed for T211R1 likely arise from conformational heterogeneity in solution. Furthermore, we now report DEER data on another new site (C599R1), which is also on the cytoplasmic side and yields high quality DEER data comparable to the S525R1 data (commended for their quality by both the reviewers). The C599R1 measurements show that in all conditions tested, highly similar distributions are observed, inconsistent with the in silico predicted distance distributions from the symmetric X-ray structures, but consistent with an asymmetric hybrid structure (i.e. open-closed) in solution. Importantly, the difference between the fully open (6.8 nm modal distance) and fully closed (4.8 nm modal distance) states of the C599R1 dimer is larger than for the S525R1 dimer pair. Thus, delineating the asymmetric hybrid conformation from the symmetric conformations is more robust.

(3) 'Our new structures, together with DEER distance measurements that monitor the conformational ensemble equilibrium of TmPPase in solution, provide further solid experimental evidence of asymmetry in gating and transitional changes upon substrate/inhibitor binding.'Problem: See above. The DEER data do not support any asymmetry.

We feel that the reviewer comments here are somewhat unfounded. All the DEER data for 525R1 periplasmic and C599R1 cytoplasmic sites are described, most parsimoniously, using an asymmetric hybrid structure. In particular, the new C599R1 distance distributions are poorly described by the symmetric X-ray crystal structures, with a conserved modal distance of approx. 5.8 nm throughout the tested conditions that aligns nicely with the in silico predictions from the asymmetric hybrid structure. Additionally, all S525R1 and C599R1 data well exceed the relevant criteria of the recent white paper (Schiemann et al., 2021, JACS) from the EPR community to be considered reliably interpretable (strong visual oscillations in the raw traces; signal-to-noise ratio .r.t modulation depth of > 20 in all cases; replicates have been performed and added into the maintext or supplementary; near quantitative labelling efficiency (evidenced by lack of free spin label signal in the CW-EPR spectra); analysed using the CDA (now Figure EV10) to avoid confirmation bias).

While the DEER data do not prove asymmetry, we do not claim proof of asymmetry in the above sentence. We concede to rephrase the offending sentence above as: “Our new structures, together with DEER distance measurements that monitor the conformational ensemble of TmPPase in solution, do not exclude asymmetry in gating and transitional changes upon substrate/inhibitor binding and are consistent with our proposed model.” We feel that this reframed conjecture of asymmetry is well founded; indeed, comparing all the 16 experimentally derived DEER distance distributions for the 525R1 and 599R1 sites with *in-silico* modelling performed on the hybridised asymmetric structure (i.e., comprised of one monomer bound to Ca2+ and another bound to IDP) yields overlap coefficients (Islam and Roux, JPC B, 2015) of >0.85. This implies the envelope of the modelled distance distribution is quantitatively inside the envelope of the experimental distance distributions. Thus, the DEER data support asymmetry (previously observed by time-resolved XRC) in solution, and while we appreciate that ideally one would measure time-resolved DEER to directly correlate kinetics of conformational changes within the ensemble to the catalytic cycle of mPPase, (and this is something we aim to do in the future), it is far beyond the scope of this study.

Indeed, half-of-the-sites reactivity has been demonstrated in at least the following papers

(Vidilaseris et al, Sci Acv. ,2019, Strauss et al, EMBO Rep. 2024, Malinen et al Prot Sci, 2022, Artukka et al Biochem J, 2018; Luoto et al, PNAS, 2013). Half-of-the sites activity requires asymmetry in the mechanism, and therefore asymmetric motions in the active site (viz 211) and exit channel (viz 525). As mentioned above, we have demonstrated this for other inhibitors (Vidilaseris et al 2019) and as part of a time-resolved experiment (Strauss et al 2024). In fact, given the wealth of evidence showing that the symmetrical crystal structures sample a non- or less-productive conformation of the protein, it would be quixotic to propose the DEER experiments - in solution - do not generate asymmetric conformations. It certainly doesn’t obey Occam’s razor of choosing the simplest possible explanation that covers the data.

(4) Based on these observations, and the DEER data for +IDP, which is consistent with an asymmetric conformation of TmPPase being present in solution, we propose five distinct models of TmPPase (Figure 7).Problem: Again, the DEER data do not support any asymmetry and the authors may revisit the proposed models.

We have redressed the proposed models and limited them to four asymmetric models to clearly illustrate the *apo*/+Ca/+Ca:ETD-state (model 1) and highlight the distinct binding patterns of various inhibitors (ETD, ZLD and IDP; model 2-4), which result in a variety of closed/open-open states. In this version, we clarify that the proposed models are not solely based on the DEER data but all DEER data recorded for multiple conditions, inhibitors and for two opposite membrane side facing reporters are highly consistent, and are grounded in both current and previously solved structures, with the DEER data providing additional consistency with these models.

(5) 'In model 2 (Figure 7), one active site is semi-closed, while the other remains open. This is supported by the distance distributions for S525R1 and T211R1 for +Ca/ETD informed by DEER, which agrees with the in silico distance predictions generated by the asymmetric TmPPase:ETD X-ray structure'Problem: Neither convincing nor supported by the data

We respectfully disagree with the reviewer. However, owing to the conformational heterogeneity of T211R1, we now exclude T211R1 data from quantitative interpretation of changes to the conformational ensemble. Instead, we include new DEER data from site C599R1, which provides high-quality and convincing data that is consistent with asymmetry at the cytoplasmic face, and inconsistent with in silico distance distributions derived from symmetric X-ray crystal structures. Furthermore, the S525R1 distance distributions for the +ETD (corresponding to +Ca/ETD) and +ZLD conditions were directly compared with both the *apo*-state distance distribution (corresponding to a fully open, symmetric conformation) and the in silico predicted distributions of the asymmetric hybrid structure (corresponding to an open-closed conformation). Overlap coefficients were calculated (given in the main text) that indicated the +ETD (corresponding to +Ca/ETD) and +ZLD S525R1 distributions were more consistent with the apo-state distance distribution. This suggests that while on the cytosolic face of the membrane, an open-closed conformation is favoured, on the periplasmic face, a symmetric open-open conformation is favoured.

**Recommendations for the authors**:
**Reviewer #1 (Recommendations for the authors):**
(1) The DEER experiments were performed with the two crystallized inhibitors, ETD and ZLD, along with previously characterized IDP. It would increase the impact of a tighter-binding phosphonate was examined since the inhibitory mechanism of these molecules is of greater interest.

We acknowledge the reviewer concern regarding the choice of weaker inhibitors. We chose to focus on the weaker binders, as we were able to obtain high-quality crystal structures for these compounds. This allowed us to perform DEER spectroscopy with the added advantage of accurately analysing the data against structural models derived from X-ray crystallography. In the revised version, we also include results from alendronate and pamidronate, two of the tighter inhibitors, which show similar and consistent results to the others.

(2) I'm not able to find the concentrations of ETD and ZLD used for the DEER experiments. This information should be added to the Methods section on sample prep for EPR.

The information is already mentioned in the Method section on sample preparation for EPR spectroscopy (page 24), where we indicated that the protein aliquots were incubated with a final concentration of 2 mM inhibitors or 10 mM CaCl2 (30 min, RT). However, we recognise that this may not have been sufficiently clear. To clarify, we now explicitly state that the concentration of ETD and ZLD (amongst other inhibitors) used for the DEER experiments is 2 mM.

(3) There should be additional detail about the electrometry replicates. Does "triplicate" mean three measurements on the same sensor, three different sensors, and different protein preparations? At a minimum, data should be collected from three different sensors to ensure that the negative results (lack of current) for ETD and ZLD are not due to a failed sensor prep. In addition, Data from the other replicates should be shown in a supplementary figure, either the traces, or in a summary figure. Are the traces shown collected on the same sensor? They could be, in principle, since the inhibitor is washed away after each perfusion.

Yes, by 'triplicate', we mean three measurements taken on the same sensor. All traces shown were collected from a single sensor. Thank you for your advice; we now show here additional data from other sensors that display the same pattern. As for the possibility of a failed sensor preparation, this is unlikely since we always ensure the sensor quality with the substrate (PPi) as a positive control after each measurement.

(4) I'm confused by the NEM modification assay, and I don't think there is enough information in this manuscript for a reader to figure out what is happening. Why is the protein active if an inhibitor is present? I understand that there is a conformational change in the presence of the inhibitor that buries a cysteine, but the inhibitor itself should diminish function, correct? Is the inhibitor removed before testing the function? In addition, it would be clearer if the cysteines that are modified are indicated in the main text. I don't understand what is being shown in Figure Ev2. Shouldn't the accessible cysteines in the apo form be shown? Finally, the sentence "IDP has been reported to prevent the NEM modification..." does not make sense to me. Should the word "by" be removed from this sentence?

We apologize for the confusion. Yes, the inhibitors were removed before testing the protein function. In Figure EV2, the accessible cysteines are shown for both the apo and IDP-bound states. As seen, the accessible cysteines in the IDP-bound states are fewer than those in the apo state, meaning fewer cysteines are available for modification. Consequently, more activity is retained when IDP binds due to the reduction in accessible cysteines. We have addressed this in the manuscript (see the method section on the NEM modification assay).

(5) Why does the model in Figure 7 show the small molecules bound to only one subunit, when they are crystallized in both subunits?

We propose that the small molecules bound to the two subunits in the crystal structure is likely a result of substrate inhibition, given the excess inhibitor used during crystallisation (e.g. Artukka, et al., Biochemical Journal, 2018; Vidilaseris, et al., Science Advances, 2022). Our PELDOR data indicate that in solution, the small molecules bound to TmPPase are in an intermediate state between both subunits being closed and both being open, most likely with at least one subunit in an open state. This is also consistent with previous kinetic studies (Anashkin, V. A., et al., International Journal of Molecular Sciences, 22, 2021), which showed that the binding constant of IDP to the second subunit is around 120 times higher than that of the first subunit.

(6) The authors argue that the two ETDs bound in the two protomers adopt distinct conformations. Can this be further supported, for example, by swapping the position of the two ETDs between the two protomers and calculating a difference map (there should be corresponding negative/positive density if the modelling of the two different conformations is robust)?

As per the reviewer suggestion, we swapped the positions of the two ETDs between the protomers and calculated the difference electron density map. This analysis, presented in Figure EV3, reveals corresponding negative and positive electron density peaks, indicating that the ETDs indeed adopt distinct conformations in each protomer, supporting the accuracy of our modeling.

(7) Are the changes in loop conformation possibly due to crystal packing differences for the two protomers?

We examined the crystal packing of the two protomers and found no interactions at the loop regions (red coloured in Author response image 2 below) that could be attributed to crystal packing differences. Therefore, we rule out this possibility.

**Author response image 2. sa3fig2:** 

(8) Typos:Legend for Figure EV2 cystine - cysteinePage 14, last sentence of the first paragraph: further - furtherFigure 6 legend: there is no reference to panel B.

Thanks for pointing out the typos, now they are fixed.

**Reviewer #2 (Recommendations for the authors):**
(1) T211 is located on the same loop where ligand/inhibitor-coordinating side chains (E217, D218) are located. It has not been tested whether spin labeling here would affect inhibitor binding.

We test all the mutant(s) activity before spin labelling, but not the activity of the spin-labelled mutants. MTSSL spin labels are typically not structurally perturbing. In particular, the T211R1 site that the reviewer is referring to is now not included in our interpretation of conformational changes occurring during mPPase’s functional cycle.

(2) Why should the spin label be introduced to T211, which is recognized as a flexible region in the crystal structure? Authors should search for suitable residues except for T211 and other residues in this loop to evaluate the cytoplasmic distance.

We acknowledge the reviewer’s concern regarding the flexibility of the T211 region for spin labelling. Given the challenges associated with TmPPase, including reduced protein expression, loss of function, or inaccessibility upon spin labelling at certain sites, we have explored alternative residues. After extensive testing, we identified C599 as a suitable site for spin labelling resulting in high-quality DEER data. The results from spin labelling at C599 have been incorporated into the revised manuscript.

(3) On the other hand, DEER data for S525 is solid, as the authors stated. This residue is located on the luminal side of the enzyme. However, the description of the luminal side structure and the comparison of symmetric/asymmetric dimer in this par are missing in the paper.

We thank the viewer for their positive assessment of the S525R1 DEER data. The data for 525 and now also for 599 spin pairs are indeed solid given the strong visual oscillation we observed particularly in such a challenging system.

We presented the periplasmic sites in the crystal structure dimer (Figure 4A), highlighting both the symmetrical region and the asymmetric model in Figure 4. In the revised version, we include additional details about this region and our rationale for labeling at position S525.

(4) The conclusion models (Figure 7) are misleading. In the crystal structure, the 5-6Loop distance between each monomer should be close given the location of the dimer interface, and the actual distance between T211 in the structure (for example, in 5lzq) is about 10A. Nevertheless, the model depicts this distance longer than S525 (40.7A in 5LZQ), which would give a false impression.

We would like to apologize for the misleading model. We have now corrected the models to ensure they are consistent with their respective regions in the crystal structures.

(5) P8 last paragraphIt is hard to imagine that in a crystal lattice, the straight inhibitor always binds to monomer A, and the neighboring monomer is always attached to a slightly tilted inhibitor, which causes asymmetry. For example, wouldn't it mean that it would first bind to one of them, which would then affect the neighboring monomer via 5-6 Loop, which would then affect its binding pose? So in this case, the inhibitor did not ARAISE asymmetry, and this is where it is misleading for readers.

We apologize for the confusion. What we intended to convey is that the first inhibitor binds to one protomer, which then affects the conformation of the neighbouring monomer, ultimately influencing its binding pose. This is required for half-of-the-sites reactivity, which is well-established in this system. This is reflected in our crystal structure, where we observed asymmetry in the loop 5-6 region and the ETD orientation between the two protomers. We have addressed this in the manuscript accordingly.

(6) P11 L4 EV10 instead of EV8?

Thanks for pointing out. We have corrected it accordingly.

(7) P11 L5 It is difficult to determine whether the peak is broad or sharp. Should be evaluated quantitatively by showing the half-value width of the peak. This may also be helpful to judge whether the peak is a mixture of two components or a single one.

We have taken this analysis out and rephrased the offending sentence. We have also added the FWHM values as the Reviewer suggested, and corresponding standard deviations for the distance distributions (under approximation as Gaussian distribution).

(8) Throughout the paper, the topology of the enzyme may be difficult to follow for readers who are not experts in this field. Please indicate the membrane plane's location or a figure's viewpoint in the caption.

We acknowledge the importance of making our figures accessible to all readers. In the revised manuscript, we have enhanced the clarity of our figures by explicitly indicating the membrane plane’s location and specifying the viewpoint in each figure caption. For example, we have added annotations such as “Top view of the superposition of chain A (cyan) and chain B (wheat), showing the relative movements (black arrow) of helices. The membrane plane is indicated by dashed lines.”

(9) Figure 2B Check the color of the helix.IDP and ETD are almost the same color, so it is difficult to see the superposition. It would be easier to understand the reading by, for example, using a lighter or transparent color set only for IDPs.

We acknowledge the reviewer concern regarding the colour similarity between the IDP and ETD in Figure 2B, which hinders clear differentiation. To enhance visual distinction, we have adjusted the colour scheme by changing the TmPPase:IDP structure colour to light blue. This modification improves the clarity of the superposition, making the structural differences more discernible.

(10) Figure 2C Check the coordination state (dotted line), there appears to be coordination between E217Cg and Mg. Also, water that is located near N492 appears to be a bit distant from Mg, why does this act as a ligand? Stereo view or view from different angles, and distance information would help the reader understand the bonding state in more detail.

Yes, we confirm that Mg^2+^ is coordinated by the oxygen atoms from both the side chain and main chain of residue E217. The water molecule near N492 is not directly coordinated with Mg^2+^ but interacts with the O5 atom of one of the phosphate groups in ETD. To enhance clarity, we have updated Figure 2C (and other related figures) to include stereo views.

(11) Figure 5A: in the Bottom view (lower left), the symmetric dimer does not look symmetric. Better to view from a 2-fold axis exactly.

We have taken this figure out entirely and instead add a direct comparison to the in silico predicted distribution from the asymmetric hybrid structure to all 16 experimental DEER distributions. We have added the symmetric and asymmetric structures to Fig. 4A and view the symmetric structure along the 2-fold axis, as suggested.

(12) Figure 5B: Indicate which data is plotted in the caption.

As mentioned above, we have taken this figure out, as we felt quantifying two overlapping populations from a single Gaussian was over-interpretation of the data, and at the suggestion of reviewer 3, we have tailored our interpretation here.

(13) Figure EV8:Because the authors discuss a lot about their conclusive model based on this data, Figure EV8 should be treated as a main figure, not a supplement. However, this reviewer has serious concerns about the measurement in this figure. Because DEER for T211 is too noisy, I don't see the point in discussing this in detail. For example, in the Ca/ETD data, there is a peak near 50A, but it would be difficult for TM5 to move away from this distance unless the protein unfolds. I do not find it meaningful to discuss using measurement results in which such an impossible distance is detected as a peak.A: Show top view as in Figure 5D: 2nd row dotted line. Regarding the in silico model that is used as a reference to compare the distance information, the distance of 40-50 A for T211 in the Ca-bound form is hard to imagine. PDB 4av6 model shows that T211 is disordered and not visible, but given the position of the TM5 helix, it does not appear to be that different from the IDR binding structure (5LZQ, 10A between two T211). The structures of in silico models are not shown in the figure, as it is only mentioned as modeled in Rossetafold. Please indicate their structures, especially focused on the relative orientation of T211 and S525 in the dimer, which would allow readers to determine the distances.

We acknowledge the reviewer’s concerns regarding Figure EV8 and the DEER data for T211R1. Upon re-evaluation, we recognize that the non-oscillating nature of the DEER data for T211R1 leads to broad distributions, indicating increased conformational dynamics, which is expected for a highly dynamic loop. Consequently, we have limited the discussion and interpretation of T211R1 in the revised manuscript and focused more on C599R1.

**Reviewer #3 (Recommendations for the authors):**
A careful interpretation of the data in view of these limitations and without directly linking to asymmetry could solve the problem of the over-interpretation of the DEER data.

We respectfully disagree with the reviewer. Please see our detailed response above.

Additional comments:(1) Did the authors use a Cys-less construct for spin labeling and DEER experiments?

We utilized a nearly Cys-less construct in which all native cysteines were mutated to serine, except for Cys183, which was retained due to its buried location and functional importance. We then introduced single cysteine mutations for spin labelling. For C599, Ser599 was reverted to cysteine.

(2) The time data for position T211R1 is too short for most cases (Figure EV8D) for a reliable distance determination. No confidence interval is given for the '+Ca' sample distance distributions.

We recorded longer time traces for two of the conditions to better assign the background. We did not use the 211R1 data to reach any conclusions regarding asymmetry, which were based on the 525R1 and the 599R1 data. We now simply include T211R1 data to indicate the high mobility observed at loop5-6. We have added the confidence interval for the +Ca condition.

(3) It is recommended to mention the 2+1 artefact obvious at the end of the DEER data.

In the methods section, we have mentioned that the “2+1” artefact present at the end of the S525R1, and T211R1 DEER data likely arises from using a 65 MHz offset, rather than an 80 MHz offset (as for the C599R1 data), which avoids significant overlap of the pump and detection pulses. We also mention in the methods section that owing to the intense “2+1” artefact, the decision was made to truncate the artefact away, to minimise the impact on data treatment. As for motivation to use the lower offset of 65 MHz, we did so to maximise the achievable signal-to-noise ratio (SNR), as particularly for the T211R1 data, the detected echo was quite weak. This was further exacerbated by the poor transverse relaxation time observed at that site.

(4) Please check the number of significant digits for all the reported values.

We have addressed the number of significant digits as requested.

(5) Please report the mean distances from DEER experiments with the standard deviation or FWHM.

We have addressed this in the revised manuscript, we report modal distances rather than the mean distances and provide the FWHM and standard deviation.